# Summary of the Research Progress on Advanced Engineering, Processes, and Process Parameters of Rare Earth Green Metallurgy

**DOI:** 10.3390/ma17153686

**Published:** 2024-07-25

**Authors:** Yingqi Li, Tingan Zhang, Zhihe Dou, Wei Xie, Chuidai Lan, Guangtao Li

**Affiliations:** Key Laboratory of Ecological Metallurgy of Multi-Metal Intergrown Ores of Ministry of Education, School of Metallurgy, Northeastern University, Shenyang 110819, China; lyq00706166@163.com (Y.L.); douzh@smm.neu.edu.cn (Z.D.); xwsx87654321@163.com (W.X.); 13407072094@163.com (C.L.); gentle9313@163.com (G.L.)

**Keywords:** rare earth, production technology, electrochemical process

## Abstract

The addition of rare earth metals to aluminum alloys can effectively improve their corrosion resistance and has been widely used in the aerospace and military industries. However, the current methods for the preparation of rare earth metals involve long processing steps, high energy consumption, and high carbon emissions, which severely constrains the development of aluminum alloys. Its output is further developed. To this end, this paper reviews mainstream rare earth production processes (precipitation methods, microemulsion methods, roasting-sulfuric acid leaching methods, electrochemical methods, solvent extraction methods, and ion exchange methods) to provide basic information for the green smelting of rare earth metals and help promote the development of green rare earth smelting. Based on the advantages and disadvantages of each process as well as recent research results, the optimal process parameters and production efficiency were summarized. Studies have concluded that the precipitation method is mostly used for the recovery of rare earth elements and related valuable metals from solid waste; the microemulsion method is mostly used for the preparation of nanosized rare earth alloys by doping; the roasting-sulfuric acid leaching method is mostly used for the treatment of raw rare earth ores; and the molten salt electrolysis method is a more specific method. This is a green and environmentally friendly production process. The results of this study can provide direction for the realization of green rare earth smelting and provide a reference for improving the existing rare earth smelting process.

## 1. Introduction

Rare earths and their compounds are widely used in aerospace equipment, medical devices, military equipment, microwaves, and the electronics industry due to their unique chemical properties [1,2]. Figure 1 is a schematic diagram of the world’s rare earth resources reserves and the world’s rare earth production in 2023. It can be seen from the diagram that China’s rare earth reserves and production are the highest in the world, accounting for 40.0% and 68.6%. The extraction, separation, and purification of rare earth compounds from ore and solid waste have attracted attention at home and abroad [3,4,5]. Currently, the precipitation method, calcination-sulfuric acid leaching method, microemulsion method, and electrolytic preparation are widely used methods for the preparation of rare earth compounds [6,7,8]. The precipitation method is widely used due to its short process flow, low labor intensity, and sufficient sources of raw materials [9,10,11]. However, this technological process is accompanied by the generation of large amounts of ammonia-nitrogen wastewater and toxic waste (NO, NO_2_) gases, which seriously constrains its further development [12,13,14]. Similarly, the environmental harm of sulfur dioxide produced by the widely used calcination-sulfuric acid process has also attracted considerable attention at home and abroad. With the increasing application of special rare earth compounds in the aerospace field, the microemulsion method has become the core for the preparation of high-purity nano rare earth compounds. However, due to the complicated process of controlling the morphology and quality of the products, only small-scale production can be realized.

At present, the great emphasis on environmental protection issues at home and abroad has put the production requirements of green metallurgy of rare earths on the agenda. The production process of rare earth compounds is often accompanied by the emission of various wastes and waste gases, so it is necessary to improve the rare earth production process and reduce the generation of waste. Reducing the emission of toxic waste substances is an important issue in rare earth green metallurgy and urgently needs to be solved. With the proposal of green rare earth metallurgy policy, the precipitation method, calcination-sulfuric acid leaching method, and microemulsion method, which are accompanied by the emission of toxic and waste gases, can hardly meet the needs of green rare earth metallurgy. The emergence of electrolysis methods combined with positive-ion membrane exchange technology has improved this problem. This process is short, does not cause pollution, does not require the addition of addenda, and consumes little energy, and it can be used to prepare rare earth compounds in an environmentally friendly, green, and scalable manner [15,16].

The rare earth green metallurgy short process reduces energy consumption and human consumption, reduces the loss of heat energy in the production process, and less emissions make improvements in the process to reduce the emissions of wastewater containing ammonia and nitrogen, sulfur-containing, and fluorine-containing gases. In the preparation of rare earth compounds by electrolysis, rare earth chloride was used as a raw material. A rare earth chloride solution was electrolyzed in an electrolytic cell with a positive-ion membrane [17]. Carbon dioxide was introduced into the cathode chamber. The rare earth cations (RE^3+^) were combined with the OH^−^ generated by electrolysis and dissolved in carbon dioxide. CO_3_^2−^ produced by water precipitates rare earth carbonate. Rare earth carbonate can be used to obtain high-purity rare earth oxides through calcination. The carbon dioxide generated by calcination can be collected by equipment, and the process returns to the electrolytic preparation stage of rare earth carbonate. A device was set up above the electrolytic tank to collect the carbon dioxide generated by electrolysis. H_2_ and Cl_2_ gases were used to prepare hydrochloric acid in a hydrochloric acid generator. This process is recyclable, green, nonpolluting, and free of emissions, the process equipment is easy to set up, and the purity of the product is guaranteed without the introduction of additives.

This paper provides a review and comparative analysis of the four methods and summarizes the process parameters, production methods, and equipment raw materials used for the four methods. The precipitation method is mostly used for the recovery and reuse of solid waste and is characterized by low temperature and simple equipment. The rare earth green metallurgical process can be achieved through the use of special reagents and an optimized precipitation process; the microemulsion method can be used to prepare nanosized rare earth compounds, and a small amount of rare earth compounds are added to the alloy compounds by the doping method. The products can be used in high-end equipment manufacturing and precision medical equipment. The roasting-sulfuric acid leaching method mainly uses rare earth ore as the raw material, such as cerium fluorocarbonate, monazite, and yttrium phosphorite. When the reaction temperature is high, leaching at an atmospheric pressure (T < 100 °C), and leaching at high-pressure conditions (T < 250 °C), a relatively large amount of acid is required to participate in the reaction. The means of process improvement are based on multistep leaching, and the means of reducing energy consumption are changing the methods of feeding, acid leaching, and other methods. This paper reviews the four current production processes of rare earth compounds, reveals the source of the raw materials and the target products suitable for each process, and discusses the feasibility of the green production process of the rare earth manufacturing industry, which is an important aspect of the green metallurgical treatment of rare earth compounds. The promotion and development of smog have provided reliable reference materials, which have laid a solid foundation for the further realization of the overall goals of short processing times, low emissions, and reduced energy consumption.

## 2. Engineering Progress in the Preparation of Rare Earth Compounds

### 2.1. Precipitation Method

Peng Wei synthesized ultrafine high-entropy rare earth silicate (HERES) powder by the molten salt method, in which the elements were uniformly distributed without segregation. A “dissolution-precipitation” mechanism was proposed to elucidate the synthesis process [18]. Masatoshi Takano recovered heavy rare earth yttrium from mixed MH-Ni battery anodes and cathode powders by chlorine gas leaching via the light rare earth double sulfate coprecipitation method [19]. Peng Wei synthesized high-entropy rare earth silicate (HERE) nanopowders by coprecipitation of rare earth hydroxides and silicic acid at the atomic scale [20]. Dongmei Zhu used heavy rare earth mother liquor to design an experimental scheme based on the response surface central combination design (CCD) method to optimize the reactive-crystallization process of rare earth carbonate and to prepare crystalline rare earth carbonate. The experiment installation is shown in Figure 2. The reactive-crystallization process was carried out in the reactor with an effective volume of 500 mL, and ammonium bicarbonate solution was uniformly fed with a peristaltic pump. The device consists of feed, agitation, condensation, and heating systems, and the reaction temperature was measured by a mercury thermometer [21]. Xiaoyu Meng chose the microbial metabolite citrate as the leaching agent to conduct a semi-industrial-scale heap leaching test of 200 tons of ionic rare earth ore (IRE-ore) to recover rare earth elements (REEs) [22]. Hiroaki Onoda selectively precipitated Nd phosphate from an iron-Nd mixed aqueous solution by adjusting the pH in two steps using NaOH [23]. Chenhao Liu synthesized a new extraction-precipitation agent: N-lauroyl sarcosine (NLSA). Mechanistic analysis revealed that the extraction-precipitation process was a cation exchange process. The specificity of the precipitation process can be attributed to the steric hindrance of the amide bond that blocks the coordination of Al (III). A novel process for the enrichment of REs in bastnaesite leachate, which can effectively remove impurities and thus effectively avoid the generation of impurity removal residues, was developed [24]. Ling He used a phosphoric acid one-step selective precipitation method to separate and recover REEs and iron from a NdFeB slurry [25]. Qianying Shi investigated the potential effect of Ca on the precipitation behavior of rare earth magnesium alloys and reported that Ca may partially replace the expensive rare earth elements in precipitation-hardened magnesium rare earth alloys [26]. Xiaoyu Meng proposed a three-step precipitation method for the efficient recovery of rare earth-citric acid (RE-Cit) complexes from (bio)leaching leachate. In the three-step precipitation process, the first step, carboxylation (pH adjustment), activated part of the coordination bonds of the RE-Cit complex; the second step, calcium ions were introduced to convert the RE-Cit complex into a ternary complex; and the third step, soluble carbonate was added to bind to the reactive sites of the rare earth and/or calcium in the ternary complex to form a precipitate [27]. Behzad Vaziri Hassas investigated the recovery of rare earths from acid mine wastewater using the NaOH and CO_2_/NaOH fractional precipitation methods. In Figure 3, CO_2_ was introduced into the collected Acid Mine Drainage (AMD), and the pH was adjusted with NaOH for 24 h stabilization, and then the solution was pumped into a high-pressure (700 kPa) filtration device for precipitation [28]. Rhan Li prepared ZnO-coated γ-Ce2S3 powder by the heterogeneous precipitation method and formed a dense protective layer of ZnO to improve its chemical and thermal stability. In addition, silica was deposited on the surface of the ZnO coating by the microemulsion process to enhance the acid resistance of the powder. The results show that dual-shell rare earth sulfide powder has good stability in coloring applications [29]. Junfeng Wang studied the effect of precipitation of low concentrations of rare earth ions on humic acid in surplus sludge. Mechanistically, rare earth ions precipitate through a series of processes, including net capture, ion exchange, adsorption, and other processes [30]. Youming Yang chose neodymium as a representative rare earth element to investigate the effect of external NaCl and CO on the metastable state. Molecular dynamics studies show that Cl^−^ provided by external NaCl occupies the coordination layer of Nd^3+^, reduces the concentration of free carbonate in the solution, and delays the formation of Nd carbonate precipitates [31]. Jing-qun Yin prepared large-grained crystalline rare earth carbonate precipitates by using ammonium bicarbonate as the precipitating agent. This proved that the crystalline rare earth carbonate was a hydrated basic carbonate or oxalate, which was not a stable intermediate carbonate during thermal decomposition [32]. Hailan Zhi used dibenzyl phosphate (DBP), diphenyl phosphate (DPP), triphenyl phosphate (TPP), diphenylphosphinic acid (DPpO), and 1,1′-binaphthyl-2,2′-dihydrogenphosphate (DHGP) for the one-step conversion of enriched rare earth sulfate to RECl_3_. The results showed that the precipitation efficiency of DBP was greater than that of DPP, DPPO, TPP, and DHGP. RE can be selectively separated from Mg and Ca by DBP [33]. Feixiong Chen’s new sulfidation-roasting-water leaching process can effectively separate rare earth elements (REEs) and impurities from spent NdFeB magnets. This process mainly involves ammonium sulfate calcination, the separation of REEs by water leaching, sodium carbonate precipitation, and calcination [34]. Qiang He studied the agitation, washing, and enrichment of calcium oxide precipitates using sodium hydroxide solution and proposed a new ion adsorption-type rare earth ore extraction process involving magnesium salt leaching-calcium oxide precipitation-sodium hydroxide stirring and washing [35]. Iga Trisnawati studied the effects of temperature, pH, and stirring speed on a multistage precipitation process and reported that Na_2_CO_3_ could effectively precipitate rare earth elements (REEs) at relatively high temperatures [36]. RA Barve synthesized rare earth-activated CaSiO_3_ by a coprecipitation method. They found that a change in the Ca-to-Si ratio led to the formation of γ-Ca_2_SiO_4_ and β-Ca_2_SiO_4_ phases, which had a significant effect on the luminescence of Ce^3+^ influence [37].

### 2.2. Microemulsion Method

Sameen Aslam prepared LiNi_0.35−y_Co_0.15_PrNd_x_Fe_2−x_ O_4_ (x = 0.0, 0.035, 0.70, 0.105, 0.140, 0.175) spinel iron by doping Pr and Nd via the microemulsion method. Oxygen has uniform saturation magnetization and nonuniform remanence magnetization [38]. Irshad Ali synthesized a series of (Tb-Mn)-doped Sr_2_Co_(2x)_Mn_x_Tb_y_Fe_(12y)_O_22_ (x = 0.0~1, y = 0.0~0.1) Y materials by the microemulsion method. type hexagonal ferrite. The effects of doping manganese at the tetrahedral position and terbium at the octahedral position were investigated [39]. Li-Qin Xiong The amine-functionalized UCNPs were prepared by a modified hydrothermal microemulsion route assisted by 6-aminohexanoic acid with an amine group content of (9.5 ± 0.8) × 10^−5^ mol/g, which not only had good dispersion in aqueous solution, but also could be further conjugated with target molecules, such as folic acid (FA), as ligands. In Figure 4, the water phase of type I microemulsion is a mixture of 6-aminocaproic acid aqueous solution and rare earth chloride aqueous solution, and the water phase of type II microemulsion is NaF aqueous solution. After mixing and stirring, the two microemulsions are sealed in an autoclave for 10 h and cooled to room temperature naturally. After adding acetone and stirring, pure UCNP powder is obtained [40]. Congying Wang prepared a series of uniform LREH (RE = Y, La, Pr, Nd, Sm, Eu, Gd, Tb, Dy, Ho, Er, Tm) nanosheets by the inverse microemulsion method. LREH nanosheets were used as a 2D support precursor to load Au nanoparticles via the deposition-precipitation method, and all the derived Au/LREO catalysts exhibited uniform nanosheet characteristics. In Figure 5, the microemulsion is composed of toluene, isopropanol, and water. The clear and uniform RE(NO_3_)_3_ microemulsion is slowly dropped into the NH_3_·H_2_O microemulsion to nucleate, and the large LREH nanosheets are formed by continuous stirring at 50 °C. Then, the Au(OH)_3_ precipitate is prepared by precipitation at 80 °C using urea as raw material, which is closely combined with LREH, and then calcined at 400 °C to obtain a Au/LREO catalyst [41]. Deepak Kumar prepared Er^3+^ Yb^3+^ codoped hexagonal NaYF_4_-converted phosphor nanocrystals by the inverse microemulsion method. The effect of different concentrations of Er^3+^ ions on the intensity of the converted luminescence after excitation by a 980 nm diode laser was studied [42]. Georgeta Predeanu used concentrated nitric acid enrichment and water washing to extract and concentrate rare earth elements (REEs) from coal bottom ash (BA) and recover valuable components [43]. Nazia Karamat synthesized pyrochlore-type Ho_2_Zr_2−x_Ge_x_O_7_(x = 0, 0. 25, 0. 5, 0. 75, and 1) using the conventional microemulsion method and investigated its structural, electrical, and dielectric properties; it was found that the Ho_2_Zr_2−x_Ge_x_O_7_ ceramics had a single-phase cubic structure [44]. ZHANG Jun used a hydrothermal microemulsion method in which a quaternary microemulsion consisting of Na_3_VO_4_/NaOH and RE(NO_3_)_3_ aqueous solution was used as the medium; the surfactant hexadecyltrimethylammonium bromide (CTAB), the cosurfactant n-hexanol and the oil phase n-heptane were used as reactants, and a hydrothermal reaction was used to synthesize YVO_4_ LRE (RE = Yb^3+^/Er^3+^, Yb^3+^/Tm^3+^) nanoparticles [45]. Wenqing Zhu used a simple W/O microemulsion method to add the cetyltrimethylammonium bromide (CTAB)/alkanol/1-octene/Sm (Sm_2_O_3_) nanoparticle precursor (Sm(OH)_3_) to an aqueous solution of NO_3_ [46]. Yadong Xu explored a method for the synthesis of layered gadolinium hydroxide (LGdH) nanoparticles by the inverse microemulsion method using oleylamine as a multifunctional agent and 1-butanol [47]. Chia-Hao Hsu studied the effect of preparation conditions on the microstructure and luminescence performance of the YAG:Eu^3+^ phosphor prepared by the microemulsion method. The local environment of the Eu^3+^ ions became more symmetrical as the calcination temperature increased. As the water content in the microemulsion system decreased, the particle size of the YAG:Eu^3+^ phosphor decreased significantly [48]. M. Mortier synthesized ultratransparent glass-ceramics in the GeO_2_-PbO-PbF_2_ system by using the heterogeneous nucleation of rare earth elements of PbF_2_. The study revealed that the optical performance increased with increasing particle size [49]. S. Janssens successfully synthesized BaMgF_4_ “mustash”-shaped nanoparticles doped with Ce^3+^, Eu^3+^, Mn^2+^, and Nd^3+^ by the inverse microemulsion method. The internal quantum efficiency was 45% at room temperature, and effective energy transfer was observed in Ce^3+^:Mn^2+^ codoped nanoparticles [50]. Yanli Wu synthesized silica-coated Gd_2_(CO_3_)_3_: Tb nanoparticles by a simple inverse microemulsion method and coating process. The particles had an average particle size of 16 nm and were dispersible in water. In vitro cell imaging of the nanoprobe showed that the nanoprobe was successfully delivered into SGC7901 gastric cancer cells in a short time and delivered to NCI-H460 lung cancer cells [51]. Guofeng Wang synthesized YF_3_:Eu^3+^ nanotube bundles by the microemulsion method. X-ray diffraction, scanning electron microscopy, and transmission electron microscopy analysis revealed that each nanotube bundle was composed of many nanowhiskers [52]. Guofeng Wang used a simple microemulsion method to synthesize 1DBaSiF_6_: Yb^3+^ (20%)/Tm^3+^ (1.2%) nanorods with a rhombic structure. Under excitation at 980 nm, the nanorods exhibited bright blue upconversion luminescence, indicating that BaSiF_6_ is a good novel host material for upconversion luminescence [53]. Guofeng Wang synthesized highly uniform and monodisperse RS octahedral nanocrystals using a simple microemulsion method and characterized them via XRD and SEM. Under excitation at 980 nm, the nanocrystals emitted a faint blue color and strong UV light [54]. Shuxing Wu prepared Bi_2_O_3_ and rare earth (La, Ce)-doped Bi_2_O_3_ visible light catalysts in Triton X-100n-hexanol-cyclohexane-water reverse microemulsion. The results showed that the photocatalytic activity of rare earth-doped Bi_2_O_3_ was greater than that of the non-dopant Bi_2_O_3_ [55]. Yan Wang synthesized Eu(DBM)_3_Phen in the water phase, the surfactant Triton X-100, the cosurfactant octanol, and the oil phase cyclohexane in water-in-oil (W/O) microemulsion (DBM)_3_Phen-embedded silica nanoparticles. The size and morphology of the nanoparticles were characterized by transmission electron microscopy (TEM). Low-temperature time-resolved emission spectroscopy showed that the Eu complex in silica nanoparticles had a longer lifetime than the pure complex [56]. Zicong Jian synthesized La-TiO_2_ and Ce-TiO_2_ nanoparticles by hydrolysis of tetrabutyl titanate in a Triton X-100/n-hexanol/cyclohexane/water reverse microemulsion. The particles were characterized by X-ray diffraction (XRD), transmission electron microscopy (TEM), Fourier transform infrared spectroscopy (FT-IR), and thermogravimetric (TG) analysis [57]. Feng Li prepared a polyvinylpyrrolidone (PVP)-stabilized Pt colloidal catalyst in a H_2_PtCl_6_ aqueous solution/PVP/n-butanol microemulsion system using hydrazine hydrate as a reducing agent. The addition of a small amount of Pr^3+^ modified PVP-Pt/Pr (Pt: Pr molar ratio of 1:0.14) further improved the catalyst activity. Using the microemulsion method [58]. Yan Wang synthesized Eu(DBM)_3_ Phen/silica nanospheres with a diameter of approximately 40 nm, which have the characteristic fluorescence of Eu^3+^ ions. SEM and TEM analyses revealed that the hybrid nanospheres had a core/shell structure with a good spherical surface. Eu(DBM)_3_ Phen was successfully encapsulated in SiO_2_ spheres as the chromophore core [59]. Li-Li Gao used cationic surfactant-hexadecyltrimethylammonium bromide (CTAB)-mediated and anionic surfactant-sodium dodecyl sulfate (SDS)-mediated microemulsions. The guided solvothermal method successfully achieved the controllable synthesis of Eu_2_(WO_4_)_3_ nanostructures with different morphologies, i.e., ellipsoidal, rod, cubic, rod bunch, and mesoporous spindle-shaped [60].

### 2.3. Roasting-Sulfuric Acid Leaching Method

Jianfei Li used a selective sulfation method to recover valuable metals from the defluorination of rare earth fluoride molten salt electrolytic slag. Based on thermodynamics and Thermogravimetric Analysis—Differential Scanning Calorimetry (TGA-DSC) analysis, the results of REEs and Li defluorination and selective sulfation were better at 500–800 °C. A flow diagram describing the selective sulfation of electrolytic slag using (NH_4_)_2_SO_4_ is shown in Figure 6. The roasting process was performed in a vacuum/atmosphere tube furnace. An alkaline solution was used for tail gas treatment. The roasting product was added to a three-necked flask (500 mL) fitted with a temperature controller with a precision of ±1 °C, and a mechanical agitator (300 rpm) was used to keep the slurry suspended [61]. Yucheng Liu used a mechanochemical method to treat REF_3_ smelter slag (RSS) generated by the catalytic reduction of REF_3_ and then leached the product with dilute acid. The results show that REF_3_ can be completely converted to a rare earth hydrate (RE(OH)_3_) by a mechanochemical method, and the REEs in the product can be efficiently leached by HCl at room temperature. As shown in Figure 7, the particle size of RSS will gradually decrease in the case of continuous action of mechanical force, and its overall specific surface area will increase progressively, while the REF_3_ embedded inside the RSS will be “exposed”, so that REF_3_ will be fully immersed in the NaOH environment. At the same time, the surface of REF_3_ particles contact and react with NaOH to form a dense RE(OH)_3_ product layer, and the REF_3_ particles will be wrapped firmly by the dense RE(OH)_3_ product layer and prevents the external NaOH from reacting with its internal REF_3_. However, under continuous mechanical force, the RE(OH)_3_ product layer was broken and re-exposed the REF_3_ in the alkaline environment. At that time, REF_3_ came into react with NaOH again, and re-generated RE(OH)_3_ product layer. In this way, the REF_3_ gradually disappears until it is fully converted to RE(OH)_3_ [62]. Shiliang Chen studied the effects of acidity, roasted concentrate particle size, temperature, and the liquid–solid ratio on the leaching kinetics of REEs, Fe, and P in the H_2_SO_4_ system to improve the selective leaching efficiency of REEs and treat solid waste [63]. Jing Zhao proposed dry treatment technology for waste gas and successfully implemented it in a rare earth roasting kiln. In this process, the dust removal, cooling, and acid condensation of the off-gas are successfully realized through the cyclone separator and the heat exchange-condensing acid system, and the waste heat is converted into steam through the steam drum, which greatly simplifies the treatment process of the off-gas torrefaction [64]. Ahmad Nawab studied a method to increase the total recovery of rare earth elements (TREEs), especially heavy rare earth elements (HREEs), from coal sources under the condition of significantly reducing the acid concentration. Three pretreatment methods for leaching were proposed, namely, (1) calcination, (2) direct acid roasting, and (3) acid roasting after calcination. Compared with simple calcination using the same amount of acid, direct acid calcination without thermal pretreatment significantly improved the recovery of TREE. The correlation coefficients of blank calcination, direct acid calcination, and second-stage acid calcination showed that there was a strong correlation between heavy rare earth elements and aluminum recovery [65]. Xingyu Liu proposed a novel closed-loop process for the recovery of rare earths and their associated resources (F and P) from Baiyun Obo rare earth concentrate via an “oxidation roasting-acid leaching → alkaline fusion → separation and purification” system. More than 98% of fluorine and phosphorus can be retained in the leaching residue by oxidation roasting-hydrochloric acid leaching. Then, the leaching residue was converted to rare earth hydroxides, sodium fluoride, and sodium phosphate in a sodium hydroxide submolten salt system. The washed filter residue was then dissolved in hydrochloric acid leaching solution to obtain a rare earth chloride solution [66]. Shiliang Chen, based on industrial operating conditions with a low liquid–solid ratio of 2:1, studied the dissolution of light rare earth sulfates in a stand-alone system at different temperatures and with different concentrations of Fe_2_(SO_4_)_3_, H_2_SO_4_, and H_3_PO_4,_ and solubility in the mixed system. For a single LREE sulfate system, the preferential crystallization of RE sulfate could be achieved by controlling the temperature and iron content [67]. Meng Wang used Mg(HCO_3_)_2_ as a precipitant to study the impact of SO^2−^ on behavioral effects. During forward feed, SO_4_^2−^ entered the La and Ce precipitates as a double salt; during synchronous feed, SO_4_^2−^ entered the precipitates as inclusions; and during the countercurrent feed, SO_4_^2−^ entered the precipitates in the form of inclusions. SO_4_^2−^ does not enter the precipitate [68]. Yanyan Zhao proposed the processes of oxidation roasting, HCl leaching, and sulfuric acid-roasting. The effect of phase change behavior during the hydrochloric acid leaching process during the oxidation roasting process at 450 °C~600 °C was investigated [69]. Dmitry Zinoveev investigated magnetic tailing-free samples obtained by carbothermal roasting of red mud at 1300 °C for 60 min and nonmagnetic tailing samples obtained by mixing 84.6% red mud and 15.4% Na_2_SO_4_ at 1150 °C by mass. The samples were subjected to high-pressure acid leaching in the range of 50 to 250 degrees Celsius at an agitation speed of 350 rpm for a duration of 30 to 90 min with 10% to 20% HCl, followed by magnetic separation of metallic iron [70]. Kamil Chadirji-Martinez studied the synthesis of Th-doped anhydrite in aqueous solution and sulfuric acid solution at different temperatures and pH values and simulated the growth of monazite-(Ce) ore under the conditions of the sulfuric acid-roasting method (SARM) [71]. Tushar Gupta investigated the use of low-temperature plasma (LTP) oxidation as a pretreatment method to improve the leaching ability of rare earth elements (REEs) from coal and its byproducts [72]. Wanyan Li proposed a multistage extraction method based on the existence of REEs in red mud. First, iron in slag is recovered and enriched in REEs by oxalic acid leaching, roasting, and dilute hydrochloric acid leaching. Subsequently, sulfuric acid leaching was used to selectively dissolve the REEs in the leachate. Finally, the effects of the sulfuric acid concentration, liquid–solid ratio, reaction temperature, and reaction time on the leaching rate of rare earths were investigated [73]. Xin-jun Bao developed a feasible method for the recovery of cerium and lanthanum from rare earth polishing powder waste to synthesize shape-controllable CeO_2_ and perovskite-type La_0.6_Ca_0.4_CoO_3_ powders. This process includes six steps: (i) preroasting treatment; (ii) H_2_SO_4_ digestion and water leaching; (iii) double salt (NaRE(SO_4_)_2_·xH_2_O) precipitation; (iv) conversion of NaRE(SO_4_)_2_·xH_2_O to RE-carbonate; (v) oxidative separation of cerium and lanthanum; and (iv) oxalate precipitation conversion and calcination to synthesize CeO_2_ and glycine-nitrate combustion process La_0.6_Ca_0.4_CoO_3_ was obtained [74]. Ghazaleh Shakiba applied deep eutectic solvents (DESs) as green substitutes for conventional solvents such as sulfuric acid and hydrochloric acid in the leaching of monazite. For the characterization of the monazite concentrate, 11 different deep eutectic solvents were synthesized and used in leaching experiments. The results showed that none of the synthesized deep eutectic solvents could extract rare earth elements from the phosphate phase [75]. Chao Kang used a step-by-step chemical extraction method to study the occurrence of Li, Ga, and REEs in coal gangue. The extraction of valuable key metals from coal gangue by the calcination activation-sulfur acid leaching process was systematically studied, and the activation mechanism was elucidated. The results show that roasting can destroy the inert kaolinite structure in coal gangue and significantly improve the leaching rate of valuable key metals [76].

### 2.4. Electrolysis and Solvent Extraction

Silvester Jürjo used praseodymium as a rare earth element to study electrochemical deposition in ionic liquids. Pr alone could not be reduced in large quantities on the Au electrode surface due to the formation of a barrier layer on the electrode surface. The addition of a small amount of bismuth salt significantly increased the deposition rate, which was very beneficial for the precipitation and separation of Pr from ionic liquids [77]. Yujian Zhou demonstrated the very effective separation of Ce^3+^ and La^3+^ by in situ electrochemical redox, which combines extraction and stripping. A new concept of combining the in situ electrolysis oxidation/extraction (IEOE) and in situ electro reduction stripping (IERS) processes was proposed. Based on the separation of REEs (Ce and La) in one step, Ce4+ from the n-octane electrolysis load was used for further reduction/stripping. In Figure 8, after leaching of the untreated spent catalyst, filtration for removal of the solid residue, and separation of the iron contaminant by solvent extraction employing diisooctyl phosphinic acid, the REE ions La^3+^ and Ce^3+^ are extracted into n-octane from the Fe-free leaching solution. The loaded organic phase (OP) containing D2EHPA/TBP is readily stripped by 0.5 M H_2_SO_4_, which can be used without further treatment in the developed CeRES process to obtain the individual REE ions in high purity in a single step without the need of additional reagents [78]. K. Hirota studied a method for the removal of oxygen from rare earth metals (Gd, Tb, Dy, Er) via electrochemical deoxidation, i.e., a titanium basket containing a rare earth metal sample was immersed in a molten CaCl electrolyte. It constitutes the cathode of the electrolysis cell, and the calcium metal generated at the cathode effectively reduces the amount of rare earth metals [79]. Y. Kamimoto investigated the electrochemical behavior of neodymium magnets during molten salt electrolysis and the effect of the composition of rare earth elements in neodymium magnets on the anodic polarization behavior and oxidation mechanism. This proved that the oxidation potential limited the oxidation stage, allowing the simultaneous leaching of rare earth elements from the hybrid neodymium magnets [80]. Prakash Venkatesan used electrochemistry to selectively extract REEs from NdFeB magnet waste at room temperature. In this electrolysis pretreatment step with NH_4_Cl as the electrolyte, NdFeB magnet waste dissolves as an active metal anode (AMA) and simultaneously, a Ti/Pt inert anode (IA) oxidizes Fe(II) to Fe(OH)_3_. Iron hydroxides have the capacity to act as metal scavengers and boron is often removed from the leachate by coagulation or electrocoagulation with ferric hydroxide. Electrolytic pretreatment is performed first to convert the elements present in the magnet waste to the corresponding hydroxides. Using a dual-anode system, NdFeB magnet scrap was used as an anode and was placed into an electrochemical reactor together with an inert anode to ensure that the iron in the magnet scrap was converted to the form of hybrid iron hydroxide; subsequently, the mixed hydroxide was leached with HCl [81]. Aarti Kumari studied the electrochemical dissolution of spent NdFeB magnets in citric acid for the recovery of rare earths and other valuable metals therein. Compared with chemical dissolution, the electrochemical dissolution of NdFeB magnets in citric acid was significantly greater. The reaction mechanism of the electrochemical dissolution was confirmed. A method was developed for the selective and quantitative extraction of rare earth elements from an electrolyte solution using di(2-ethylhexyl) phosphoric acid (D2EHPA) as the extracting agent [82]. Hanwen Chung studied the feasibility of extracting rare earth metals from magnet regenerated oxides (MRDOs) using molten salt electrolysis. MRDOs were prepared by the oxidation of waste NdFeB magnets in air, followed by carbothermal reduction under an 80 mbar argon atmosphere [83]. Xiujuan Feng used electrochemical impedance spectroscopy (EIS) to reveal the resistance change characteristics and leaching rate of rare earth elements during the in-place leaching process, established an equivalent circuit model for the leaching process, and analyzed the solution resistance Rs and charge-transfer resistance Rt. These two key parameters reflect the electrochemical characteristics. The leaching process was divided into four stages: wetting stage, reaction stage, equilibrium stage, and top water stage. Rs and Rt were negatively correlated with the pore size and chemical reaction rate during the leaching process [84]. E. Bourbos reported that at low temperature, N-butyl-N-methylpyrrolidine bistrifluoroformimide (BMPTFSI) and trimethylbutylammonium bistrifluoroformimide Me_3_NBuTFSI could be synthesized with the successful electrodeposition of rare earth metals [85]. Bradley R studied the electrochemical separation of the liquid metals, lanthanum and yttrium, by using La_2_O_3_ and Y_2_O_3_ as sesquioxides at temperatures higher than 2500 K in a molten mixture. The thermodynamic selectivity of molten rare earth oxides and their alloys deviating from the ideal mixture is several orders of magnitude greater than that of the standard state. For the first time, iridium wire electrolytic decomposition of La_2_O_3_, Y_2_O_3_, and their mixtures was used, and the electrolytic properties and ability to extract rare earth elements from these melts were confirmed [86]. Jesús R. Pérez-Cardona’s study used an electrochemical method to recover rare earth metals using RTILs as the electrolyte solution. The LCA and TEA results of RTIL showed that the environmental and economic performance of this process were comparable to those of the MSE process. In terms of environmental performance, except for ozone layer depletion, all environmental impact categories can be improved. From an economic point of view, the RTIL process is quite competitive with the MSE process [87]. Chuchai Sronsri performed acid extraction and selective recovery of rare earth elements from electronic waste under anaerobic conditions. Column extraction is used to increase the contact between the leachate solution and the solid waste. The adsorption curve was fitted to the Langmuir model and followed first-order kinetics [88]. Aida Abbasalizadeh used iron as an anode in the electrolysis process to promote the dissolution of iron into the melt, prevent the precipitation of perfluorocarbon (PFC) gases at the anode, and use the iron fluoride formed by the dissolution of iron to convert rare earth oxides into rare earth fluorine compounds. In the extraction of Nd by electrowinning, the selection of a halide, the generation of CO/CO_2_ and PFC gases, and the low solubility of NdO in molten fluoride are solved by using an anode and oxidation to fluoride [89]. Yujian Zhou introduced an effective method for the recovery and separation of Ce^3+^ and La^3+^ from spent catalytic cracking (FCC) catalysts. Both rare earth elements were back-extracted from the organic phase using H_2_SO_4_, and then the separation of Ce^3+^ and La^3+^ was achieved through in situ electrochemical oxidation and simultaneous solvent extraction. During this process, Ce^3+^ was electrochemically oxidized to Ce^4+^ and extracted from the aqueous phase with 100 mM D2EHPA in n-octane [90]. Xuan Xu used a green and convenient Nd-Fe-BPM recovery route to show that REEs and Fe metal can be recovered in two steps through electrolysis and selective precipitation [91]. S. Vasudevan studied an electrochemical pathway for the electrolytic oxidation precipitation of rare earth chloride solution to separate cerium as cerium hydroxide. The rotation of the cathode is also used to avoid the formation of hydroxide scale on the surface of the cathode [92]. Prakash Venkatesan demonstrated the effective recovery of NdFeB magnet scrap by a room temperature electrochemical process. First, the magnet scrap is completely leached with HCl, followed by in situ electrochemical oxidation to selectively oxidize Fe (II) in the leachate to Fe (III). Finally, oxalic acid was directly added to the electro-oxidation leachate, resulting in the selective precipitation of more than 98% of the REEs as RE oxalates [93]. Xiaoyan Ji used rare earth oxychlorides as precursors to prepare rare earth alloys in molten CaCl_2_ by solid cathode electrolysis, and electrochemically reduced a mixture of LaOCl and NiO to LaNi_5_. It was concluded that LaOCl neither absorbs moisture in the air nor reacts with the molten electrolyte [94]. Xiang Peng reviewed the progress in structure engineering of rare earth-based perovskite OERs in recent years. The strategies for enhancing the electrochemical properties of rare earth perovskite materials via structural engineering (including A-site substitution, B-site substitution, composite engineering, and morphology) are discussed and organized [95]. Kouji Yasuda established the recycling process of scrap neodymium magnets by the electrochemical separation of Nd, Dy, and Pr in a NaCl-KCl-RECl_3_ molten salt system at 973 K by using the RE-Ni alloy formation reaction [96].

## 3. Process Comparison of the Preparation of Rare Earth Compounds by Engineering Methods

The preparation of rare earth compounds by the precipitation method involves simple equipment, readily available raw materials, and simple process flow, which has unique advantages in the treatment of solid waste and the recovery of rare earth and other metals (such as iron, manganese, etc.). However, this process generates a large amount of ammonia-nitrogen wastewater, emits toxic gases (NO_2_ and NO), and has a relatively low purity. To solve these problems, the control of reaction conditions (such as temperature, ingredient list, reaction time, stirring speed, pH, etc.) and the optimization of process flow (such as fractional precipitation, optimization of precipitant, etc.) can be used to improve product purity and reduce industrial toxicity. Energy consumption should be reduced, and the generation of waste gas, waste material, and liquid should be reduced. Through the efforts of experts and scholars, this process can now obtain micron-sized products with a purity of more than 95%. Table 1 lists the comparison and analysis of the process parameters for the preparation of rare earth compounds by the precipitation method, which integrates the research methods and results of the past ten years.

Microemulsion methods can be used to manufacture nanosized rare earth elements and their compounds. In most cases, the doping method is used to add a small amount of rare earth elements such as samarium, neodymium, europium, and dysprosium to the alloys or compounds to be prepared to improve or enhance the performance of product properties. The magnetic properties, luminescence performance, catalytic performance, and stability of the prepared alloys and compounds are greatly enhanced. However, this process has high equipment accuracy, accurate reaction operation, and strict condition control requirements and is unsuitable for large-scale scale-up, which limits the further development of this process. Table 2 lists the comparison and analysis of the process parameters for the preparation of rare earth compounds by the microemulsion method and summarizes the progress made by all parties in the microemulsion method over the past 15 years.

The roasting-sulfuric acid leaching method is currently the mainstream process for the large-scale processing of rare earth ore and the preparation of rare earth compounds in industry around the world, and it has the advantages of a high level of industrialization, complete process flow, and large processing capacity. However, a higher temperature is required in the calcination stage of this process, and a large amount of sulfuric acid is used in the leaching stage, which generates various sulfur-containing waste gases and waste liquids, causing environmental pollution. To solve related problems, domestic and international scholars have adjusted process parameters (such as temperature, solid-to-liquid ratio, pH, reaction time, stirring rate, etc.) and improved process methods (such as material addition methods, advanced acid leaching, ore crushing methods, etc.) to reduce the energy consumption of the process, reduce the generation of waste liquid and gas, and improve the product purity. Table 3 shows the process parameters for the preparation of rare earth compounds by calcination-sulfuric acid leaching showing the comparison and analysis of the research progress on this process over the past ten years.

Rare earth compounds with relatively high purities were prepared from solid waste and rare earth fluorides. This process is characterized by simple equipment, low labor intensity, and high product purity. It can extract and separate rare earth elements from solid waste very well, and the product purity can reach more than 99%. Using the existing process flow of the chlor-alkali industry as a reference, the research group established a process flow for the preparation of rare earth compounds by electrolysis. This process is short, does not cause pollution or low energy consumption, and can provide a good basis for rare earth green metallurgical projects. An exemplary blueprint for. Table 4 lists the parameters for the preparation of rare earths and their compounds by means of electrolysis.

## 4. Conclusions and Prospects

The development of methods for the preparation of rare earth elements and their compounds is related to the progress of the national economy and the national defense force. The greening of production process engineering is a strong guarantee for promoting further progress in the rare earth industry. This paper focused on the various production process parameters of rare earth elements and their compounds and compared four processes—precipitation, microemulsion, roasting-sulfuric acid leaching, and electrolysis and revealed that engineering parameters (such as temperature, pH, stirring speed, and reaction time) and improving production processes (such as material feeding methods, ore crushing methods, raw material pretreatment methods, and multistep precipitation processes) can improve production efficiency, reduce production energy consumption, and reduce waste and gas. The authors concluded that the precipitation method is suitable for the recovery of rare earth elements and related valuable metals from solid waste, the microemulsion method can be used for the preparation of nanosized rare earth alloys by doping, and the roasting-sulfuric acid leaching method is mostly used for the treatment of raw rare earth ores. Electrolysis might become a green and environmentally friendly production process. This method has broader development prospects and provides a reference for the green and efficient development of industrial production at home and abroad.

## Figures and Tables

**Figure 1 materials-17-03686-f001:**
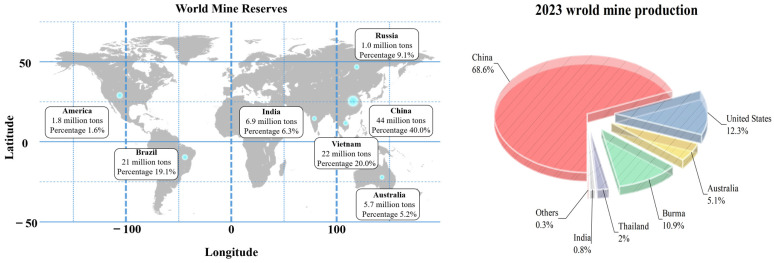
World rare earth reserves and 2023 world rare earth production.

**Figure 2 materials-17-03686-f002:**
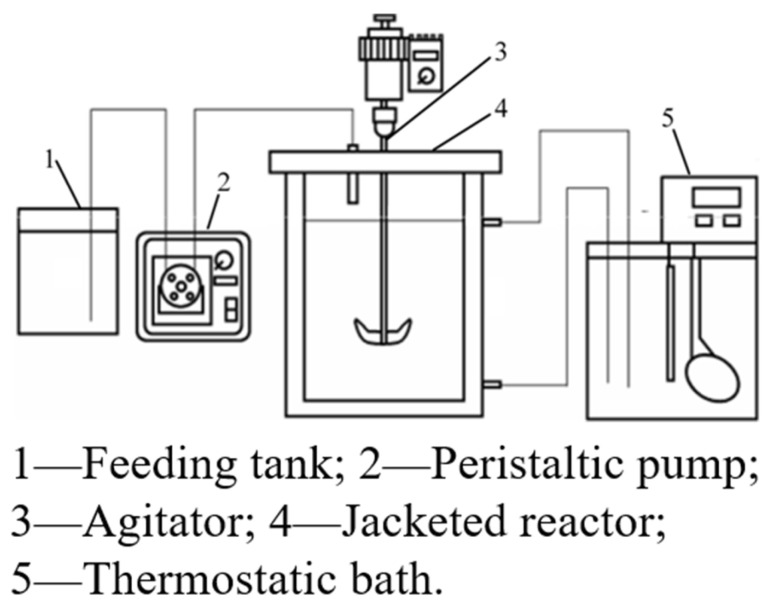
Experiment installation of the reactive-crystallization process [21].

**Figure 3 materials-17-03686-f003:**
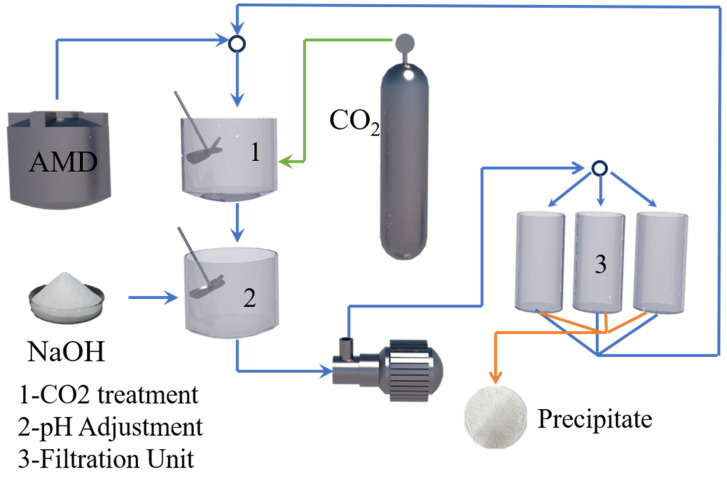
Schematic representation of the experimental setup [28].

**Figure 4 materials-17-03686-f004:**
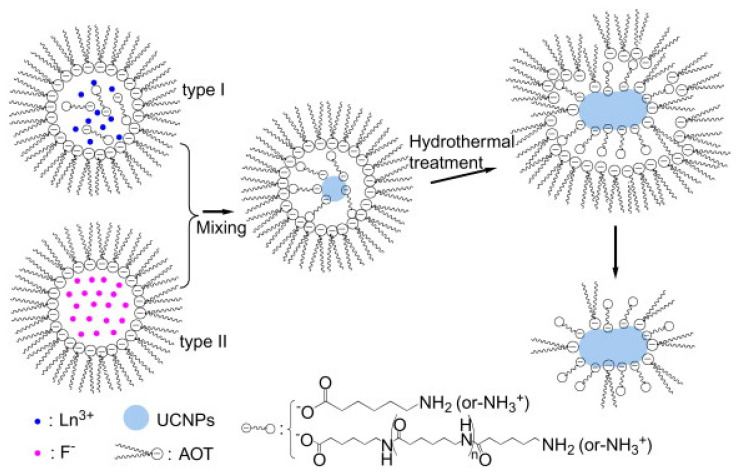
Synthesis mechanism of biocompatible UCNPs by a modified hydrothermal microemulsion route [40].

**Figure 5 materials-17-03686-f005:**
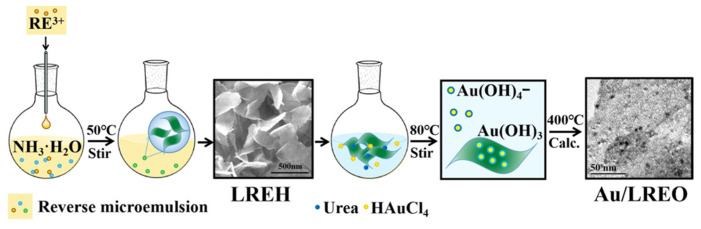
Schematic illustration for the preparation process of Au/LREO catalysts [41].

**Figure 6 materials-17-03686-f006:**
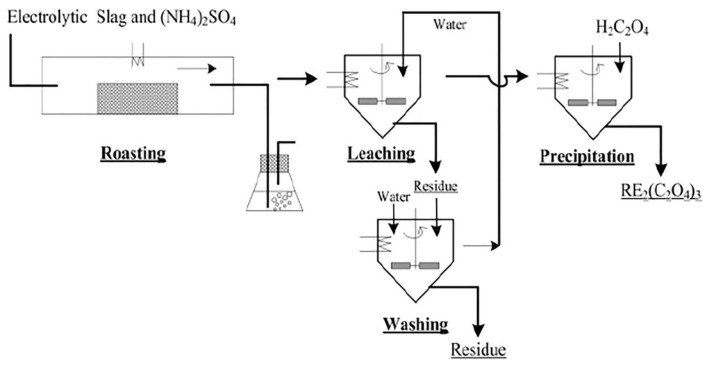
Schematic of procedure to treat the electrolytic slag [61].

**Figure 7 materials-17-03686-f007:**
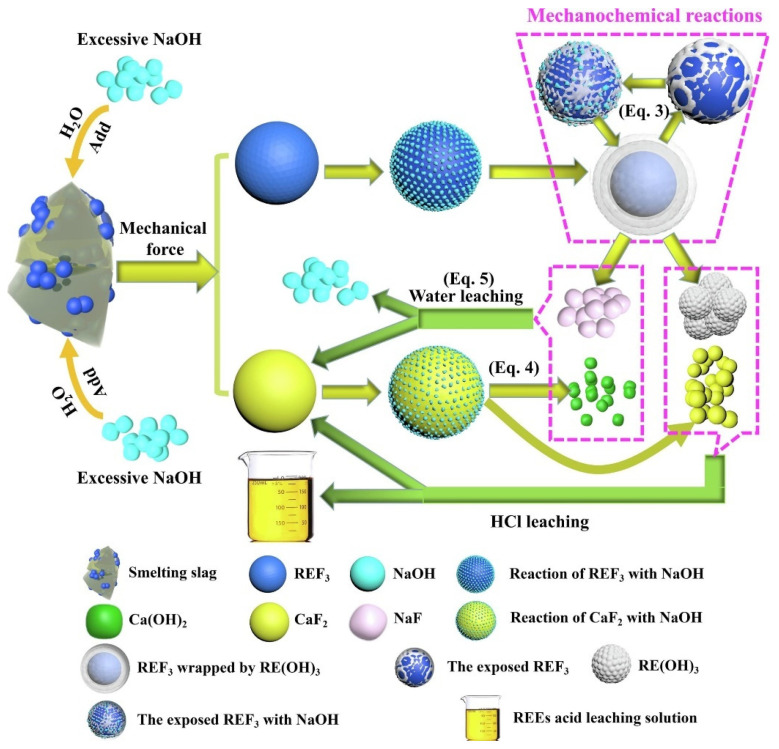
Schematic diagram of the mechanism of extraction of REEs and recovery of by-products via mechanochemical process and [62].

**Figure 8 materials-17-03686-f008:**
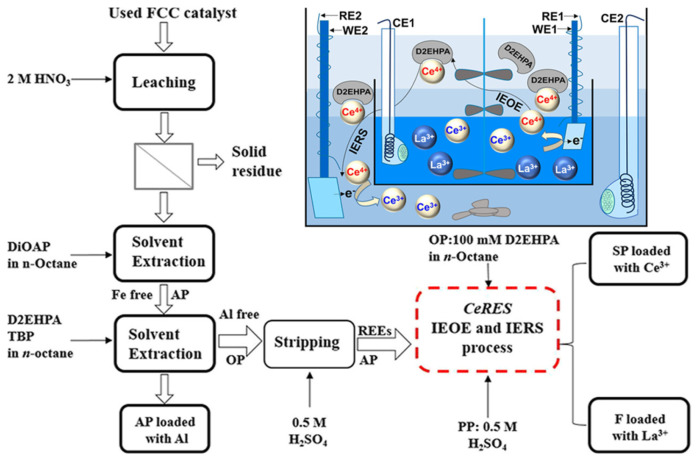
Overall protocol for the recovery of REEs from spent FCC catalysts incorporating the CeRES process with a concentric cell (OP = organic phase; AP = aqueous phase; SP = stripping phase; and F = feed phase) [78].

**Table 1 materials-17-03686-t001:** Comparison and analysis of the process parameters for the preparation of rare earth compounds by the precipitation method.

Preparation Methods	Prepared Products	Reaction Temperature	Raw Material	Reaction Conditions	Product Characteristics	Yield/ Recovery
Molten salt precipitation method [18]	HEREDs	1073 K	RE_2_O_3_:SiO_2_ = 1:2.4	2 h	Grain size: 1.32 μm	
Molten salt precipitation method [18]	HEREMs	1173 K	RE_2_O_3_:SiO_2_ = 1:0.9	3 h	Grain size; 1.64 μm	
Bisulfate coprecipitation method [19]	RE_2_O_3_	50–90 °C	Solid waste	650 rpm, 1 h	RE_2_O_3_	La > 99%, Ce > 99%
Chemical precipitation method [20]	HEREs	1173 K	RE/Si = 1:0.4	pH = 7~8	Type X2 monosilicic acid	-
Chemical precipitation method [21]	RE_2_(CO_3_)_3_	Room temperature	Heavy rare earth	Concentration: 1.75 g/L, seed crystal: 13.56 wt%, maturing: 8 h	28.23 μm, Water rhombic structure	97.82%
Organic precipitation method [22]	REEs	Room temperature	Citrate:50 mmol/L, L:S = 1:2	Ph5	RE_2_O_3_	96%
Two-step precipitation method [23]	NdPO_4_	Room temperature	0.002 mol/LNdCl_3_·6H_2_O, NaOH, 0.2 mol/LH_3_PO_4_	pH 4	NdPO_4_	99.97%
Phosphoric acid one-step precipitation method [25]	REEs	80 °C	4 mol/LH_3_PO_4_, S:L = 30:1	Extraction: 1.5 h	RE_2_O_3_	99.49%
CO_2_ mineralization and precipitation method [28]	REEs	Room temperature	CO_2_/NaOH	Step 1 pH 5 Step 2 pH 7	RE_2_(CO_3_)_3_	95%
Heterogeneous precipitation method [29]	γ-Ce_2_S_3_	Room temperature	γ-Ce_2_S_3_, CTAB, Zn(NO_3_)_2_	50 °C, agitation: 5 h, calcination: 200°C	γ-Ce_2_S_3_ powder coated with ZnO	-
Bioleaching three-step precipitation method [27]	RE_2_(CO_3_)_3_	Room temperature	CaCO_3_, Na_2_CO_3_	pH > 2	RE_2_(CO_3_)_3_	96%
Sludge-based humic acid precipitation method [30]	RE_2_(CO_3_)_3_	25 °C	Low concentration rare earth solution	L:S = 1:1, 100 rpm, pH 8.6	Amorphous particle, No crystal characteristic peaks	89%
Extraction-precipitation method [24]	RE (III)	60 °C	NLSA, rare earth solution	6 mol/L HCl	61.5 μm	96%
Chemical precipitation method [31]	Nd	Room temperature	NaCl, CO	0.5 mol/L NaCl, 450 min	RE_2_(CO_3_)_3_	93%
Chemical precipitation method [32]	RE_2_(CO_3_)_3_	40–70 °C	NH_4_HCO_3_	pH 5–8, 500 rpm	50~200 μm	95%
Chemical precipitation method [33]	RECl_3_	Room temperature	DBP, DPP, TPP, DPPO	pH 4.5	300 μm	99.99%
Enrichment and precipitation method [35]	RE_2_O_3_	35 °C	n(NaOH)/n(RE^3+^) = 2.85, L:S = 6.5 mL/g	Agitation: 20 min	5 μm	94.389%
Multistage precipitation method [36]	LREEs, HREEs	50 °C	RE^3+^, oxalic acid, NaOH	15%Na_2_CO_3_, pH 4.5, 200 rpm	10 μm	88%/74%
Coprecipitation method [37]	γ-Ca_2_SiO_4_, β-Ca_2_SiO_4_	1200 °C	RE, Ca, Si	n(Ce) = 1, 1.43, 2	Fluorescence lifetime: 33.5, 34.3, 36.5 ns	-

**Table 2 materials-17-03686-t002:** Comparison and analysis of the process parameters for the preparation of rare earth compounds by the microemulsion method.

Prepared Products	Process Means	Means of Characterization	Product Characteristics	Product Performance
LiNi_0.35−y_Co_0.15_Pr_y_Nd_x_Fe_2−x_O_4_ [38]	Doping method	XRD, FTIR, VSM	28~70 nm	Magnetic(al) matrix performance: −10,000~10,000 Oe
Sr_2_Co_2x_Mn_x_Tb_y_Fe_12y_O_22_ [39]	Doping method	SEM, VDXS, VSM	Y type hexagonal ferrite	Magnetic(al) matrix performance > 3200 Oe
UCNPs [40]	Hydrothermal method	LSUCLM, XRD, EDXA, FTIR	20~40 nm	Amino amount: (9.5 ± 0.8) × 10^−5^ mol/g
LREH nanosheets [41]	Inversion method	XRD, SEM, FTIR	Homogeneous nanosheets	Dimension < 3 nm)
NaYF_4_ fluorescent nanocrystalline powder [42]	Inversion method	XRD, TEM, SEM	Hexagonal nanopowder	a = b = 5.9168 Å, c = 3.331 Å
Enrichment and concentration of REE [43]	Concentrated nitric acid enrichment and water washing	XRD, SEM	Extract	-
Ho_2_Zr_2−x_Ge_x_O_7_ [44]	Conventional law	XRD, SEM	Single-phase cubic structure	Dielectric loss < 1.3 GHz
YVO_4_: RE (RE = Yb^3+^/Er^3+^, Yb^3+^/Tm^3+^) [45]	Hydrothermal method	XRD, TEM	Nanoparticles	Controllable particle size, narrow particle size distribution, and less agglomeration
Sm(OH)_3_ [46]	W/O type	DSC-TGA, XRD, TEM, UV–VIS	Nanoparticles	Nanoparticles
Layered gadolinium hydroxide (LGdH) [47]	Inversion method	XRD, TEM, FTIR	200 nm	Size-controlled nanoparticles
YAG: Eu^3+^ phosphor powder [48]	W/O type	XRD, TEM	100 nm	Good particle size and good luminosity
Glass-ceramic [49]	Doping method	XRD, TEM, DTA	20 nm	Strong optical performance
BaMgF_4_:Mn^2+^:Ce^3+^ nanoparticles [50]	Doping method	XRD, TEM, EDS	50~80 nm	Internal electronic efficiency = 45%
Silica-coated Gd_2_(CO_3_)_3_:Tb nanoparticles [51]	Inversion method	HRTEM, EDS, FTIR	16 nm	This nanoprobe was successfully delivered to gastric cancer SGC7901 cells in a short period of time and delivered to NCl-H460 lung cancer cells
YF_3_:Eu^3+^ nanobeams [52]	Doping method	XRD, SEM, TEM	Length = 500 nm, diameter = 2 nm	High luminous intensity
BaSiF_6_:Yb^3+^ (20%)/Tm^3+^ (1.2%) [53]	Doping method	XED, TEM	Length = 1 mm	High luminous intensity
YF_3_:Yb^3+^ (20%)/Tm^3+^ (2%) [54]	Doping method	XRD, SEM	100 nm	Good luminosity
Bi_2_O_3_ doped RE catalyst [55]	Doping method	XRD, TEM, BET	Nano monoclinic crystal	High catalytic ability
Eu(DBM)_3_ [56]	W/O type	TEM, Olympus optical camera	40 nm	Regular shape, high photobleaching resistance
Nano-La-TiO_2_ and Ce-TiO_2_ particles [57]	Inversion method	XRD, TEM, FTIR, TG	20~50 nm	Good photocatalytic activity
Polyvinylpyrrolidone (PVP) stabilized Pt colloidal catalyst [58]	Catalytic reflux method	TEM, XPS	Nanoparticles	High catalytic activity
Eu(DBM)_3_Phen/SiO_2_ nanosphere [59]	Doping method	SEM, TEM	40 nm	good spherical surface
Eu_2_(WO_4_)_3_ nanocrystalline [60]	Solvothermal method	XRD, TEM	Nanoparticles	Controllable morphology
Mn_1−x_Ni_x_Fe_2−y_Dy_y_O_4_ soft magnetic oxide [97]	Doping method	XRD, FTIR, VSM	20~30 nm	Increase in saturation magnetization

**Table 3 materials-17-03686-t003:** Comparison and analysis of the process parameters for the preparation of rare earth compounds by calcination and sulfuric acid leaching.

Process Means	Means of Characterization	Process Conditions	Reaction Material	Yield/Recovery/Conclusion
Selective sulfation method [61]	TGA-DSC, XRD, SEM	(NH_4_)_2_SO_4_/slag = 3:1, 750 °C, 60 min	REF_3_ 1.0MLiOH	REEs > 96.5%
Echanized method [62]	XRD, SEM	NaOH/RSS = 0.4:1, 400 rpm, 40 min	REF_3_	REEs > 96%
Sulfuric acid leaching method [63]	ICP–AES, XRD, SEM	15 min, 40 °C, 250 rpm, L/S = 7:1	REF_3_	REEs > 97%
Exhaust gas dry process [64]	XRD	Adding cyclone separator and heat exchange-condensing acid system	REEs	H_2_SO_4_ > 91.17% fluoride > 93.44%
Acid-roasting method [65]	ICP–OES, XRD, TGA-DSC	600 °C, 50 g/L, 2 h	Coal	REEs > 80%
Oxidation roasting-acid leaching [66]	ICP–AES, TGA-DSC, SEM	Calcination: 550 °C, 2 h acid pickling 8 MHCl, 20 min, 70 °C	Rare earth concentrate	REO > 280 g/L
Acid leaching [67]	ICP–AES	25~65 °C, L:S = 2:1 Fe_2_(SO_4_)_3_ (0~50.13 g/L), H_3_PO_4_ (20.34 g/L) 0.5 M H_2_SO_4_	Rare earth sulfate	Independent system is higher than hybrid system
Forward addition method [68]	TGA-DSC, XRD, SEM	n(RE^3+^): n(HCO^3+^) = 1:3, 30°C, 300 rpm	Rare earth sulfate	Forward feeding improves efficiency
Oxidation roasting-acid leaching [69]	TGA-DSC, XRD, SEM	4 M HCl, 25 °C, 300 rpm, 1 h, L:S = 6:1	Astnaesite	REEs > 70.32%
Carbonate thermal reduction method [70]	XRD, SEM	150 °C, acid concentration: 10%, L:S = 1:11, 60 min	Red mud	REEs > 98%
Sulfuric acid-roasting method [71]	XRD, SEM, ICP-MS	353~473 K, pH = 0.3~3	Monazite mine	Th^4+^ > 1780 mg/L
LTP oxidation method [72]	XRD, SEM	600 °C, 5 h, L:S = 1.6:1	Coal	HREE > 53%
Multistage extraction acid leaching [73]	TGA-DSC, XRD, SEM	1 M H_2_SO_4_, 3 h, 95 °C, L:S = 5:1	Red mud	REEs > 80%
Roasting-acid leaching [74]	TGA-DSC, XRD, SEM	Six-stage treatment	Rare earth solid waste	REEs > 91.73%
Acid leaching [75]	XRD, SEM	500 °C, 2 h, S:L = 10:1	Monazite	HREE > 94%
Step-by-step chemical extraction method [76]	TGA-DSC, XRD, SEM	Calcination conditions: 650 °C, 2 h; leaching conditions: 120 °C, acid concentration 6 M, L/S = 10:1, 1 h	Coal gangue	Rees = 344 μg/g

**Table 4 materials-17-03686-t004:** Comparison and analysis of the process parameters for the preparation of rare earth compounds by electrolysis.

Purpose	Process Means	Means of Characterization	Experimental Conditions	Conclusion
Study on the electrochemical deposition of Pr [77]	Addition of bismuth ions	XRD	Cyclic voltammetry curve	The addition of bismuth ions is more conducive to the precipitation of Pr
Separation of La/Ce [78]	In situ electrochemical redox for extraction and back-extraction	Autolab PGSTAT 100 N potentiostatic/galvan-ostatic	Electrolysis 60 min, −1.0 mA/cm^2^	La^3+^ > 99.7%
Removing oxygen from RE [79]	Electrochemical oxidation method	Electrochemical workstation	1189 k, 10 h	Reduction of rare earth metals with oxygen content greater than 2000 mass ppm to 10~50 mass ppm
Recycled neodymium magnet [80]	Potential electrolysis	XRD, SEM	−1.8~0.8 V	Recyclable
Recovery of REEs from NdFeB magnet scrap [81]	Electrolysis	XRD	500 rpm, 8 h, 25 °C, I_NdFeB_ = 0.5 A, I_Ti/Pt_ = 0.2 A	REEs > 97%
Recovery of REEs from waste NdFeB magnet [82]	Acid-dissolved electrolysis	XRD, SEM, TG-DTA	1 M D2EHPA, oxalic acid, 1073 K	REO > 99.9%
MRDO extraction of REEs [83]	Low potential potentiostatic deposition	XRD, SEM, ICP-OES	1050 °C	REEs > 98%
Electrodeposition behavior [85]	Potentiostatic deposition	SEM, EDS	−3.1 V, 25 °C, 5 h	REEs > 98%
Electrochemical separation of La/Y [86]	Potentiostatic deposition	SEM	2.5 V, 2573 K	Successful isolation
E-waste recovery of REEs [88]	Potentiostatic deposition	Langmuir model	160 A/m^2^, 7 mL/min, n(Fe^3+^) = 0.8 mol/L	Recovery rate =1.135 mg/min
FCC recovery La^3+^/Ce^3+^ [90]	In situ electrochemical oxidation	LSV, CV, SWV	25% D2EHPA, 25% TBP	La^3+^ > 99.5%, Ce^4+^ = 100%
Electrodeposition behavior [89]	Potentiostatic deposition	CV	RECl_3_	REO > 95%
Recovery of REEs from NdFeB magnet scrap [91]	In situ electrochemical oxidation	XRD, CV	Oxalic acid precipitation	REO > 99.2%
Recovery of REEs from waste magnets [93]	Potentiostatic deposition	SEM	973 k, 0.42 V	Effective separation of Nd and Pr

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
