# Peer review of "Summary of the Research Progress on Advanced Engineering, Processes, and Process Parameters of Rare Earth Green Metallurgy"

_materials, 2024, doi:10.3390/ma17153686_

Round 1
Reviewer 1 Report
Comments and Suggestions for Authors
You may find review report in the attachement, the main reason for rejecting the submitted manuscript is given in the text below:
Checking the full papers of numerous references such as 3-35 and even more (I didn’t find it necessary to check all of them) I came to conclussion that those cited references can’t be brought in connection with the corresponding text in submitted manuscript.
Thanks to references irrelevance to the text in submitted manuscript I came to decission that this manuscript should be rejected.

The English throughout the text is fine, only some minor flaws can be found, most of them are just lapsus calami.
Author Response
Manuscript Materials-3089342
- Response to Reviewers
Dear Editor and Reviewers,
Thank you for giving us the opportunity to submit a revised draft of the manuscript “Summary of the research progress on advanced engineering, processes and process parameters of rare earth green metallurgy” for publication in the Materials. We appreciate the time and effort that you and the reviewers dedicated to providing feedback on our manuscript and are grateful for the insightful comments on and valuable improvements to our paper.
- Comments from the Editors and Reviewers:
- Reviewer 1:
You may find review report in the attachement, the main reason for rejecting the submitted manuscript is given in the text below:
Checking the full papers of numerous references such as 3-35 and even more (I didn’t find it necessary to check all of them) I came to conclussion that those cited references can’t be brought in connection with the corresponding text in submitted manuscript.
Thanks to references irrelevance to the text in submitted manuscript I came to decission that this manuscript should be rejected:
- Author response:
Thank you very much for the time and effort that the reviewers put into the manuscript, and we summarized two questions from the reviewers' suggestions. Firstly, the overall comprehensiveness and influence of the article are explained. Secondly, we explain the correspondence between the literature (3 ~ 35) cited in the manuscript and the article. As the reviewer proposed, the overall comprehensiveness and influence of the manuscript have good characteristics. However, due to the author 's negligence, the strong correlation between the cited literature and the article was not specified, which caused a misunderstanding that the literature did not correspond to the manuscript. We will explain in detail through the instructions below and mark the changes in the article in blue font. All changes do not affect the main conclusions of the article. Finally, thanks again for the reviewer 's suggestions on the manuscript, which can not only improve the reader 's intuitive reading experience, but also play an important role in improving the quality of the manuscript. Below is our detailed answer to the suggestion, hope to make you satisfied.
- The overall comprehensiveness and influence of the manuscript
- The novelty of this paper mainly comes from three aspects:
- firstly, it summarizes four methods used in the field of production of rare earths and their compounds in the past ten to twenty years, and summarizes and evaluates each method.
- advocates the integration of greening, short processes, and cost reduction into the field of rare earth industrial production. Targeted description of the advantages of the electroconversion method for the preparation of rare earth compounds and its production process.
- Reasons for each innovation point
- Firstly, four methods used in the field of production of rare earths and their compounds in the last decade to two decades are summarized and each method is summarized and evaluated.
Before writing this manuscript, we investigated the rare-earth production processes at home and abroad in the past ten years according to the requirements of carbon reduction and carbon neutrality put forward by the international community, and reviewed a large number of related literature, and found that although the rare-earth industrial production processes have been developing greatly, they still cannot get rid of the large amount of toxic waste gases and liquids discharged in the process of production, which is hazardous to the protection of the environment due to its high energy consumption. In order to show the advantages and disadvantages of each method more clearly, we summarize the four mainstream production processes, such as precipitation, microemulsion, roasting-sulfuric acid leaching, electrolysis, etc., and analyze the research content of scholars, and summarize four tables (see Table 1, Table 2, Table 3, Table 4 in the manuscript), which introduces dozens of specific methods of production of raw materials, process parameters, means of characterization, products, properties and so on. properties, etc. According to these four tables, we can find domestic and foreign experts innovating in multiple areas, such as improving production process parameters and simplifying the production process, in order to achieve the purpose of increasing production efficiency, reducing production and consumption, and reducing emissions.
- Advocating the integration of greening, short process and cost reduction into the field of rare earth industrial production. Targeted description of the advantages of electrochemical conversion method for the preparation of rare earth compounds and its production process.
This manuscript is written to emphasize the importance of greening production for the development of advanced manufacturing processes for rare earths. In order to achieve the goals of short flow, low energy consumption and reduced emissions in the production process, we propose a production process means for the preparation of rare earth compounds by electrolysis. The process takes the chlorinated rare earth solution as raw material, utilizes the cation membrane's property of partitioning the solution and releasing the cations to pass through, and transfers the rare earth cations to the cathode of the electrolysis chamber to form a precipitate under the condition of applying an electric field. The process has a production process without the introduction of impurities to ensure the purity of the product, the precipitation process for the ion co-deposition effect, the product grains for the micron to nano-scale, the production process only consumes electricity, able to wind power, solar power and other forms of power, more in line with the demand for green production process.
We hope that you will find the above innovative and informative. We have made changes to the manuscript based on your comments, and we have listed the reasons and explanations for these changes below. We are grateful for the time and effort you have put into the manuscript and for the specialized questions you have asked. The manuscript was made more professional by working together.
- Correlation between references and manuscripts
Reference 3 titled: Radioactive Main Group and Rare Earth Metals for Imaging and Therapy, described at P912, lines25 “Scandium-44 has garnered attention as a PET radionuclide due to its high β+ branching ratio (Eβ+avg=632 keV, 94%) and long physical half-life (t1/2 = 4.04 h). The latter is advantageous for several reasons, most notably the potential to acquire images at later time points than common β+ emitters, such as 18F(t1/2=1.83h) and 68Ga(t1/2=1.13h). Background signals typically decrease with time due to progressive radiotracer clearance; thus, delayed imaging with 44Sc may improve tumor-to-tissue ratios and lead to higher image quality. Imaging later time points also presents the opportunity to investigate pharmacokinetics of slower-circulating bioconjugates. Lastly, the extended half-life may permit cost-effective use of 44Sc through centralized production and regional distribution, which would not be feasible with a shorter-lived radionuclide. Scandium-44 is considered a diagnostic match for the common β− emitters 90Y and 177Lu, due to the chemical similarities of rare-earth metals. More appealing, however, is the prospect of combined use with 47Sc due to their shared chemical identity. Scandium-47 (t1/2=80.4 h) is a high branching, low-energy β− emitter (Eβ−avg=162 keV, 100%) suitable for treatment of small tumors and cancer metastasis. β− emission is accompanied by coemission of low-energy γ-rays (Eγ=159 keV; Iγ=68%) useful for SPECT imaging. Unfortunately, regular clinical use of 47Sc is a distant prospect due to inconvenient production methodology. Other useful radioisotopes for scandium-based radiopharmaceuticals include the β+ emitter 43Sc (t1/2=3.89 h, Eβ+avg=476 keV, 88%), which is free of high-energy gamma emission, unlike 44Sc (Eγ=1157 keV; Iγ=100%) and 44mSc (t1/2=58.6 h), due to its potential as an in vivo 44Sc generator.” 。The preparation of rare earth element Sc and its application in the medical field are pointed out, in which the part of the application of rare earth is strongly correlated with the part of the manuscript “The extraction, separation and purification of rare earth compounds from ore and solid waste have attracted attention at home and abroad.”
Reference 4 titled: Rare Earth Metal-Mediated Precision Polymerization of Vinylphosphonates and Conjugated Nitrogen-Containing Vinyl Monomers is described in P2001, line19. lines19 describes “REM-GTP of DAVP proceeds via an SN2-type associative displacement of the polymer phosphonate ester by a vinylphosphonate monomer with a pentacoordinated intermediate (Scheme 9). The first transition state, i.e., the monomer coordination, represents the rate-determining step. In this transition state, the metal-(O=P) bond to the activated monomer is much longer than the metal−(O−P) bond to the polymer ester. The longer Ln-(O=P) bond leads to a relatively small steric demand of the added vinylphosphonate in comparison with the growing chain end. The resulting minor effect of the steric demand of the added monomer on the propagation rate is in accordance with previous observations (vide supra), thus providing evidence for the proposed existence of a pentacoordinated intermediate.” Described in P2001, lines34 “Smaller metal centers destabilize the propagation ground state by a more confined arrangement of the eight-membered metallacycle according to the higher steric constraints caused by shorter Ln−Cp, Ln−(O−P), and Ln-(O=P) bonds. The destabilization of the ground state is not enthalpic (i.e., ring strain or the Ln-(O=P) bond strength), but entropic in nature (i.e., rotational and vibrational limitations in the eight-membered metallacycle). In the transition state, the Ln-(O=P) polymer phosphonate ester bond is lengthened, thus compensating for part of the steric stress induced by the coordination of a vinylphosphonate monomer. This effect is larger for a stronger destabilization of the ground state, i.e., for smaller metal centers.” 。The application of lanthanides in the formation of coordination sites with vinyl phosphonates is noted, where the section on rare earth applications has a strong correlation with the manuscript, “The extraction, separation and purification of rare earth compounds from ore and solid waste have attracted attention at home and abroad.”
Reference 5 titled: Rare Earth Starting Materials and Methodologies for Synthetic Chemistry, described in P6044, lines7 “Eu and Yb ammonia solutions are relatively stable when stored under anaerobic conditions. However, they decompose over time to generate the parent amide Ln(NH2)2, similar to what is observed with alkali and AE metals (Scheme 2). In the case of Yb, depending on reaction conditions, degradation of ammoniacal solutions can also produce oxidized trivalent amide Yb(NH2)3. Divalent Eu(NH2)2 is obtained also under ammonothermal reaction conditions (up to 5000 atm of NH3), while for Yb the same reaction conditions generate pure trivalent Yb amide or the salt Na[Yb(NH2)4]. The use of ammonothermal conditions with Lns has limited synthetic utility and has attracted interest mostly for the preparation of new semiconductors; also, supercritical ammonia (160 °C) has been used to prepare metal sulfide salts of Yb, [Yb(NH3)8]-[M(S4)2]·NH3 (M=Cu, Ag), and La, [La(NH3)9][Cu(S4)2]. Additionally, Müller-Buschbaum and Quitmann reported the molecular structure of complex [Sm(NH3)9][Sm(Pyr)6] (Pyr=pyrrolide, {C4H4N}−), obtained from the direct reaction of pyrrole with Sm metal and pyrrole under solvothermal conditions, followed by treatment with liquid ammonia. Recently, Kraus and co-workers investigated the preparation of Eu(II), Yb(II), and Ho(III) azides by reacting the pure metal with AgN3 in liquid ammonia, leading to the structural identification of ammino-adducts [Ho2(μ-NH2)3(NH3)10]-(N3)3·(NH3)1.25 and [Yb(NH3)8](N3)2.”。 The applications of Eu and Yb in the production of semiconductors are noted, and the section on rare earth applications has a strong correlation with the manuscript, “The extraction, separation and purification of rare earth compounds from ore and solid waste have attracted attention at home and abroad.”
Reference 6 titled: Rare-Earth Doping in Nanostructured Inorganic Materials, described in P5519, lines1 “Impurity doping is a promising method to impart new properties to various materials. Due to their unique optical, magnetic, and electrical properties, rare-earth ions have been extensively explored as active dopants in inorganic crystal lattices since the 18th century. Rare-earth doping can alter the crystallographic phase, morphology, and size, leading to tunable optical responses of doped nanomaterials. Moreover, rare-earth doping can control the ultimate electronic and catalytic performance of doped nanomaterials in a tunable and scalable manner, enabling significant improvements in energy harvesting and conversion. A better understanding of the critical role of rare-earth doping is a prerequisite for the development of an extensive repertoire of functional nanomaterials for practical applications. In this review, we highlight recent advances in rare-earth doping in inorganic nanomaterials and the associated applications in many fields. This review covers the key criteria for rare-earth doping, including basic electronic structures, lattice environments, and doping strategies, as well as fundamental design principles that enhance the electrical, optical, catalytic, and magnetic properties of the material. We also discuss future research directions and challenges in controlling rare-earth doping for new applications.”。 It is pointed out that rare earth doping is applied to the application of material properties, and the preparation of rare earth doped materials has a strong correlation with the “ methods for the preparation of rare earth compounds.” in the manuscript.
Reference 7 Title: Recent Development in Sensitizers for Lanthanide-Doped Upconversion Luminescence, in P16001, lines4 Description “The straightforward impact of UCL using Yb3+ as sensitizer is that at least several orders of magnitude of luminescence greater than that in singly doped crystals could be obtained through optimization of the doping concentration. Apparently it would be a more efficient option to consider other rare earth ions acting as sensitizers, which may or may not be identical to the activator ions, so that the activator ions can luminesce more easily and more effectively. Later experimental results have further proved thatsuch UCL through sequential energy transfer is so efficient that it could be achieved through blackbody excitation or spontaneous diode emission and has thus become a pervading phenomenon in all rare earth-doped materials under high-density infrared excitation shortly after laser sources became commonly available. Besides the above-mentioned examples, a large amount of hosts with different crystallographic parameters were meanwhile adopted for UCL studies, such as Y2O3, YbPO4, BaYF5, NaLnF4 (Ln = Sc, Yb, Nd, Gd, etc.), NaYbF4, NaScF4, and so on. Up to now, UCL from lanthanide activators of Tb3+ (4f8 ), Ho3+(4f10), Tm3+(4f12), Tm2+(4f13), Pr3+(4f2), Nd3+(4f3), Gd3+(4f7), Dy3+(4f9), Sm3+(4f5), Sm2+(4f6), Eu3+(4f6), Eu2+(4f7), transition metal ions (TMn+) including but not limited to Ti2+(3d2), Cr3+(3d3), Ni2+(3d8),Bi3+(6s2), Mn4+(3d3), Mn2+(3d5), Re4+(5d3), Mo3+(4d3), and Os4+(5d4), and even actinides such as U3+(5f3) and U4+(5f2) was subsequently reported.”。 It is pointed out that Yb is used as a sensitizer to make light-emitting diodes, and the preparation of rare earth doped materials has a strong correlation with methods for the preparation of rare earth compounds.” in the manuscript.
Reference 8 Title: The Coming of Age of Neodymium: Redefining the Role in Rare Earth Doped Nanoparticles, in P518, lines9 Description “Among the most promising of such theranostic NPs are materials composed of rare earth (RE3+) ions-rare earth doped NPs or RENPs. RE3+ ions that participate in light absorption and emission (lanthanoids) each have a unique energy level configuration, which enables easy spectral and temporal discrimination of their luminescence bands in the UV, visible, and NIR spectral ranges. Furthermore, RENPs are chemically stable and are not prone to photo-oxidation because their luminescence arisesfrom the 4f electronsthat are not involved in chemical bond formation. RENPs can be synthesized through soft chemistry techniques with functional groups that can be easily modified and then conjugated with biological molecules, making them excellent candidates for sensing, bioimaging, and photoactivated therapies. RENPs are considered nontoxic (though more research is still needed to fully support this statement). Most notably, the RENPs are nowadays playing a key role in the shift toward NIR luminescence imaging, particularly relevant for biomedical purposes.”。 It is pointed out that rare earth ions are used as materials to prepare therapeutic NPs, and the preparation of rare earth doped materials has a strong correlation with 'methods for the preparation of rare earth compounds. ' in the manuscript.
Reference 9 Title: YAG:Ce3+ Phosphor : From Micron-Sized Workhorse for General Lighting to a Bright Future on the Nanoscale, in P13461, lines4 Description “The renowned yellow phosphor yttrium aluminum garnet (YAG) doped with trivalent cerium has found its way into applications in many forms: as powder of micron sized crystals, as a ceramic, and even as a single crystal. However, additional technological advancement requires providing this material in new form factors, especially in terms of particle size. Where many materials have been developed on the nanoscale with excellent optical properties (e.g., semiconductor quantum dots, perovskite nanocrystals, and rare earth doped phosphors), it is surprising that the development of nanocrystalline YAG:Ce is not as mature as for these other materials. Control over size and shape is still in its infancy, and optical properties are not yet at the same level as other materials on the nanoscale, even though YAG:Ce microcrystalline materials exceed the performance of most other materials. This review highlights developments in synthesis methods and mechanisms and gives an overview of the state of the art morphologies, particle sizes, and optical properties of YAG:Ce on the nanoscale.”。 It is pointed out that the convenience of rare earth material preparation, in which the short process of rare earth manufacturing method has a strong correlation with the ' short process flow, low labor intensity, and sufficient sources of raw materials ' in the manuscript.
References 10 Titles: Nano-and micro-sized rare-earth carbonates and their use as precursors and substrates for the synthesis of new innovative materials, P2035, lines21 Description “The different REOHCO3 polymorphs have different coordination types. The coordination number of RE in the orthorhombic Pnma REOHCO3 structure is 10 (Fig. 1). The RE atom is coordinated by ten O atoms from two hydroxyl groups and five carbonate groups, where three of the carbonate groups are coordinated to the rare-earth atom as chelate ligands and the other two are coordinated as monodentate ligands. In the orthorhombic REOHCO3 with a P212121 space group the coordination number of the RE atom decreases to 9 as one of the chelate ligands changes to a monodentate ligand (Fig. 1). In the hexagonal REOHCO3 the coordination number of the RE atom is also 9. However, in this case the RE atom is coordinated to six oxygen atoms of four carbonate groups, of which two are chelate ligands and two are monodentate ligands. The coordination sphere is completed by three oxygen atoms from three hydroxyl groups. In the tetragonal structure the coordination number of the RE atom is the lowest among the REOHCO3 structures (Fig. 2a). In this polymorph the RE atom is coordinated to six oxygen atoms of four carbonate groups – two are chelate ligands and two are monodentate ligands. Additionally two hydroxyl groups coordinate to the RE atom. Therefore the RE atom in the tetragonal phase has a coordination number of 8. Only one crystal structure of a rare-earth oxycarbonate hydrate compound has been reported – an orthorhombic La2O(CO3)2H2O with a Pmcn space group. In this crystal structure the La atom is coordinated by 10 O atoms (Fig. 3). Three of the carbonate groups are coordinated to the rare-earth atom as chelate ligands and two are coordinated as monodentate ligands. Additionally two hydroxyl groups coordinate to the La atom. The rare-earth dioxycarbonates can crystallize in three different structures: tetragonal (I), monoclinic (Ia), and hexagonal (II). All three polymorphs are believed to hold an arrangement of RE2O2 layers separated by CO32- ions.”。 The coordination number and crystal composition of rare earth carbonates are pointed out. The analysis of rare earth carbonates has a strong correlation with the manuscript ' The precipitation method is widely used due to its short process flow, low labor intensity, and sufficient sources of raw materials. '
References 11 Title : Exploring heavy fermions from macroscopic to microscopic length scales, in P6, lines6 Description “Inelastic neutron scattering experiments performed on the quantum critical material CeCu6−xAux (xc=0.1) revealed a scaling of the dynamical susceptibility as a function of the energytemperature ratio (E/T, FIG.5a) — a similar observation had been previously made for UPd5−xCux (REF.96). In CeCu6−xAux, the critical temperature and energy exponent is fractional, α=0.75. The same value of α describes the magnetic susceptibility at wave vectors far away from the ordering wave vector of the AF phase at x>xc (FIG.5b). These observations have prompted the development of new theoretical concepts in heavy-fermion quantum criticality. In the local QCP scenario (FIG.3b), strong low-dimensional order-parameter fluctuations are assumed to break the composite heavy charge carriers. Theory predicts that this destruction of the Kondo effect is accompanied by a discontinuous change of the Fermisurface volume from large to small. This has been explored for CeRhIn5 in the low‑T normal state — that is, in magnetic fields of 10–17T, which suppress superconductivity — through direct Fermi-surface measurements performed using de‑Haas–van-Alphen oscillations of the magnetization for different pressures across the AF QCP at pc≈2.3GPa (REF.86). A pronounced jump in the Fermi-surface volume occurs at the QCP (FIG.5c; as T approaches zero, p=pc). In addition, a clear tendency towards the divergence of the cyclotron mass, mc, is observed on either side of pc (FIG.5d). All of these observations are clear-cut signatures of a Kondo-destroying QCP in pressurized CeRhIn5.”。 Description in P10, lines18 “Ce3+, Sm3+ and Yb3+ are Kramers ions and frequently exhibit a magnetic doublet crystal-field‑derived ground state. The Kondo coupling to the conduction electrons introduces a finite lifetime for the crystal-field states of the localized 4f electrons, and the lowest-lying crystal-field states can only be separated from the excited ones if the Kondo coupling is sufficiently weak. This is visualized by a double-peak structure in ρ(T) and in the temperature dependence of the thermoelectric power, S(T), in which the peaks at elevated temperatures (at TKhigh) indicate the onset of the Kondo screening of the fully degenerate Hund’s rule ground state (j=5/2 for Ce3+ and Sm3+, and j = 7/2 for Yb3+), including all crystal-field states, which are populated according to Boltzmann statistics. By contrast, the position of the low‑T maximum indicates the Kondo temperature, TKlow, of the crystal-field‑derived lowest-lying Kramers doublet; this temperature is commonly denoted simply as TK. Data for CeCu2Si2 and its 20% La‑substituted variant are shown in FIG.10a. In the stoichiometric compound, Tcoh (~15–20K) is very close to TK (as obtained from entropy considerations). The bizarre temperature dependence of the thermoelectric power of this compound is well described by the scattering term in the Mott formula for the thermoelectric power, S(T) (FIG.10a), which can be written as N(T)/μH(T)150, where the Nernst coefficient, N(T), and the Hall mobility, μH(T)151, are related to the on‑site conduction-electron/4f‑electron Kondo and skew-scattering processes, respectively. This implies that in this Kondo-lattice system, the local Kondo screening is dominant down to temperatures of the order of ~0.1Tcoh. Therefore, the standard description of the low-T thermoelectric power in heavy fermions, in terms of the asymmetric energy dependence of the heavy-fermion DOS at the Fermi level, is not adequate in this temperature regime, even though it may be appropriate at very low temperatures, at which the renormalized band structure is almost completely developed.”。 It is pointed out that the magnetic application of rare earth ions, in which the application of rare earth ions has a strong correlation with ' short process flow, low labor intensity, and sufficient sources of raw materials ' in manuscripts.
References 12 Title : Hydrophobicity of rare-earth oxide ceramics, in P315, lines3 Description “Hydrophobic materials that are robust to harsh environments are needed in a broad range of applications. Although durable materials such as metals and ceramics, which are generally hydrophilic, can be rendered hydrophobic by polymeric modifiers, these deteriorate in harsh environments. Here we show that a class of ceramics comprising the entire lanthanide oxide series, ranging from ceria to lutecia, is intrinsically hydrophobic. We attribute their hydrophobicity to their unique electronic structure, which inhibits hydrogen bonding with interfacial water molecules. We also show with surface-energy measurements that polar interactions are minimized at these surfaces and with Fourier transform infrared/grazing-angle attenuated total reflection that interfacial water molecules are oriented in the hydrophobic hydration structure. Moreover, we demonstrate that these ceramic materials promote dropwise condensation, repel impinging water droplets, and sustain hydrophobicity even after exposure to harsh environments. Rare-earth oxide ceramics should find widespread applicability as robust hydrophobic surfaces. ”。It is pointed out that the manufacture of rare earth ceramic materials will have a bad impact on the environment, and the environmental pollution caused by it has a strong correlation with ' which seriously restricts its further development. ' in the manuscript.
References 13 Title : Low-oxygen rare earth steels, described in P1137, lines11 “Research and development of RE elements and RE steels have been undertaken in metallurgical fields for many years. The first introduction of RE materials into steels was tracked back to the 1920s in the United States; subsequently, RE steels have been investigated worldwide. These studies have demonstrated the highly attractive advantages of RE addition to steels. For instance, the addition of RE elements (for example, La and Ce mischmetal) into steel melts (1) exhibits effective purification during deoxidization and desulfurization, (2) allows modification of inclusions and (3) enables micro-alloying. Occasionally, these effects lead to an improvement of the material properties, including toughness, plasticity, heat resistance, corrosion resistance and wear resistance. With the discoveries of these effects, RE addition to steels has become an important technology, leading to its extensive application in industry. In the 1980s, the RE steel yield in the United States was claimed to be as high as nine million tons per year, approximately one-tenth of the total annual steel production in the United States; at that time, the main effects of RE addition were deoxidization and desulfurization. RE steels have been a special focus in China because of the rich RE resources there. RE steels have been produced in China since the 1950s, and in the late 1980s, the number of related studies peaked.”。 It is pointed out that rare earth will help remove oxygen and sulfur from cast iron in steelmaking, and desulfurization will cause environmental pollution, which has a strong correlation with 'which seriously restricts its further development. ' in the manuscript.
Reference 14 Title : Multipole polaron in the devil 's staircase of CeSb, in P413, lines25 Description “It is important to find a relationship between the evolution of the multipole polaron and the devil’s staircase. The AFP magnetostructure is a consequence of the competing CEF ground states (Fig. 1b): the ferromagnetic f Γ80 plane and the paramagnetic f Γ7 plane. The presence of such competing states can often lead to a spatial modulation of the electronic interaction. Indeed the modulation of the p–f mixing strength was previously considered in CeSb as an origin for the AFP superstructure, where the p–f mixing strength is periodically suppressed, leaving the paramagnetic f Γ7 planes (Fig. 1b). The observation of no considerable change of λ at the transition from the AFP3 to AFP5 phase (Fig. 4i) indicates that the multipole polaron is insensitive to the change of the f Γ7 superstructures. By sharp contrast, an abrupt enhancement of the multipole polaron occurs at the transition from the AFP5 to AFP6/AFF phase. Notably, this enhancement coincides with the strong specific heat jump reflecting the appearance of the AFF phase, where all paramagnetic layers disappear in the whole crystal (the black dashed line in Fig. 4i). If the AFF magnetostructure is considered, the enhancement of λ likely occurs when all the paramagnetic f Γ7 planes are eliminated in the crystal and only the f Γ80 superstructure (q=6/11) is left (Fig. 1b). Furthermore, the large λ changes when the f Γ80 superstructure of the AFF phase (q=6/11) is transformed into the simple double-layer modulation (q=1/2) of the AF phase. The unusual temperature evolution of λ sensitively varying with the elimination of the f Γ7 planes and the modulation of the f Γ80 superstructure is consistent with our picture of the multipole polaron that is strongly coupled to the f Γ80-to- f Γ∗8 excitation under ferromagnetic f Γ80ordering. Also, such a correlation between the multipole polaron and the f Γ80 ordering is likely related to the pseudo-gap-like behaviour previously observed at the temperatures corresponding to the AFP6/AFF phase. In addition, a sharp reduction of the resistivity across each transition was previously observed; the conductivity becomes better with increasing (reducing) the number of the f Γ80 (f Γ7) planes. Thus, our observation of the coherent QP state developing in the f Γ80 ordering corresponds to these resistivity changes.”。 It is pointed out that the manufacture of CeSb multipolaron will produce pollution, and the pollution environment has a strong correlation with 'which seriously restricts its further development. ' in the manuscript.
References 29 Titles : Technique for Enhanced Rare Earth Separation, described in P2327, lines14 “The chemical potentials of chlorine corresponding to Ln+LnCl2 and LnCl2+LnCl3 equilibria at 1073 K are shown in Fig. 1. The Gibbs energies of formation of the trichlorides (LnCl3) that are available in the literature are combined with those for the dichlorides (LnCl2), which are estimated from their enthalpies of formation, For unstable LnCl2 compounds, the enthalpies of formation evaluated by Novikov and Polyachenok and Johnson using the Born-Haber cycle were used. These data, associated with larger uncertainty, are represented by dotted symbols in Fig. 1. When pure LnCl2 is stable, the chemical potential of chlorine corresponding to Ln+LnCl2 equilibrium is more negative than that for the LnCl2+LnCl3 equilibrium. This is the case for the elements Nd, Sm, Eu, Dy, Ho, Tm, and Yb. For the other elements, the pure dichloride is unstable, and hence, the chemical potential for the Ln+LnCl2 equilibrium is more positive than that for the LnCl2+LnCl3 equilibrium. In these cases, some LnCl2 can exist at reduced activity in molten salts. It is seen that the trend in the chemical potential of chlorine for the Ln+LnCl2 equilibria is opposite to that for LnCl2+LnCl3 equilibria: minima in the first approximately correspond to maxima in the second and vice versa. Therefore, it is possible to selectively reduce a trivalent ion in a mixture of rare earth trichlorides to its divalent state. Thus, a separation process combining selective reduction with vacuum distillation is feasible. This idea is demonstrated for the binary systems Pr-Nd and Nd-Sm. The rare earth metals Nd and Sm are the most important for producing strong magnets. The combination of Pr and Nd is one of the most difficult to separate by the conventional process, requiring more than 20 repetitive extractions.”。 It is pointed out that rare earth chlorides can carry out the conversion of divalent and trivalent states, in which the valence state change and the manuscript ' This process is short, does not cause pollution, does not require the addition of addenda, and contributes little energy. and it can be used to prepare rare earth compounds in an environmentally friendly, green and scalable manner '.
Reference 31 : Mechanochemical-Assisted Leaching of Lamp Phosphors : A Green Engineering Approach for Rare-Earth Recovery, in P398, lines16 Description “Rare-earth elements (REEs) are essential metals for the design and development of sustainable energy-related applications such as renewable energy technologies (e.g., solar, wind, and thermoelectric converters), lighting, and magnetic materials. Recycling critical REEs from end-of-life consumer goods (e.g., permanent magnets, lamp phosphors, and Ni-metal hydride (MH) batteries) and recovering these elements from industrial residues (e.g., bauxite residue and phosphogypsum) are the most prominent pathways to ensure an independent supply for future applications, aside from primary mining. Recycling has major advantages over primary supply, including a smaller environmental footprint, shorter lead times, and cheaper sources of material. Moreover, recycling may provide a solution for the over-supply of ‘‘unwanted” REEs, and it can contribute to a steady supply of the more critical REEs (neodymium (Nd), dysprosium (Dy), and terbium (Tb)), thus mitigating the so-called ‘‘balance problem”.”。It is pointed out that the method of green recovery and production of rare earth elements has a strong correlation with the manuscript 'Green depositing of rare earth elements is needed to achieve the goals of short processing, low energy consumption and low rare earth emission. '
According to the suggestions put forward by the reviewers, we found that although references 15,16,17,18,19,20,21,22,23,24,25,26,27,28,30,32,33,34,35 are related to manuscripts in rare earth background, the intensity of correlation is weaker than that of other references. We adopt the reviewer 's suggestion to remove references 15, 16, 17, 18, 19, 20, 21, 22, 23, 24, 25, 26, 27, 28, 30, 32, 33, 34, 35. Thank you again for the reviewer 's suggestion. Through the modification of this article, readers can get a stronger and more intuitive experience of the industry background.
Reviewer 2 Report
Comments and Suggestions for Authors
In the manuscript, the authors presented the review of mainstream rare earth metals production processes. The work provides basic information for the green smelting of rare earth metals and help promote the development of green smelting.
Rare earths metals and their compounds are widely used in aerospace equipment, medical devices, military equipment, and the electronics industry due to their unique chemical properties. The work is relevant because it can serve as a reference for improving an environmentally friendly process for the production of rare earths metals. The great emphasis on environmental protection issues has put the production requirements of green metallurgy of rare earths on the agenda. Based on the advantages and disadvantages of each process as well as recent research results, the optimal parameters of each proces and production efficiency were summarized. Therefore, the work may be of interest to the scientific community.
The manuscript is well organized. The text is written in a way that the reader can understand. Presented methods are properly discussed and explained to the reader.
The paper provides a review and comparative analysis of the four current production processes of rare earth compounds, and summarizes the parameters, methods, and equipment used for the four groups of methods: precipitation methods, microemulsion methods, roasting-sulfuric acid leaching methods, and electrochemical methods. The authors discuss the sources of the raw material, the target products suitable for each process, the feasibility of the green production process of the rare earth manufacturing industry. Particularly useful for use by readers of the presented article are tabular comparisons and analyses of the parameters of the processes for preparation of rare earth compounds by individual engineering methods.
References contain 115 items, including the latest from the last 2 years, that are used to scientifically document and discuss the methods reviewed. The references are appropriate. There is no inappropriate self-citations by authors.
Considering the above-mentioned merits of the manuscript, I believe that the work is worth publishing.
Before publishing, the revision is required to improve the quality of the manuscript.
My specific comments:
1. Not all readers of the article will be specialists in the field, so some specific research methods may not be familiar to them. Therefore, the methods should be written in the manuscript with their full names, not abbreviated. So please give the full name "TGA-DSC analysis" in line 282.
2. According to the publisher's recommendations to the authors of the manuscript, abbreviations and acronyms should be explained the first time they are used in the text. So please explain all the acronyms, in particular:
- LTP in line 347,
- IEOE in line 384,
- IERS in line 385,
- OP in line 390
Author Response
Manuscript Materials-3089342
- Response to Reviewers
Dear Editor and Reviewers,
Thank you for giving us the opportunity to submit a revised draft of the manuscript “Summary of the research progress on advanced engineering, processes and process parameters of rare earth green metallurgy” for publication in the Materials. We appreciate the time and effort that you and the reviewers dedicated to providing feedback on our manuscript and are grateful for the insightful comments on and valuable improvements to our paper.
Reviewer 2:
We are very grateful to the reviewers for recognizing our manuscript as “contributing to the advancement of green smelting” and “The references contain 115 items, including the most recent items from the last 2 years, which were used to scientifically document and discuss the methods reviewed. The references are appropriate. There are no inappropriate self-citations by the authors.” We are very pleased with the comments, “Considering the strengths of the manuscript as described above, I believe this work is worthy of publication.” We are also grateful for the time and effort the reviewers put into the manuscript, and their suggestions were very sound, and we have revised the manuscript accordingly, as indicated by the blue font. The following is a point-by-point response to the reviewers' suggestions, which we hope you will find satisfactory.
- Not all readers of articles are experts in the field, and therefore they may not be familiar with some specific research methods. Therefore, these methods should be given their full names in the manuscript rather than abbreviations. Therefore, please give the full name “TGA-DSC analysis” on line 282.
The reviewer's suggestion is correct, and we will refer to the full name as “TGA-DSC analysis”. It is given in line 282. For your convenience, we have listed the changes below.
Based on thermodynamics and Thermogravimetric Analysis - Differential Scanning Calorimetry (TGA-DSC) analysis, the results of REEs and Li defluorination and selective sulfation were better at 500-800 °C.
- In accordance with the publisher's recommendations to authors of manuscripts, abbreviations and acronyms should be explained the first time they are used in the text. Therefore, please explain all acronyms, in particular:
The suggestions made by the reviewers were very correct and we have explained all the acronyms. They are located in lines 347, 384, 385, and 390 of the revised manuscript. For your convenience, the changes are listed below.
Tushar Gupta investigated the use of Low-temperature plasma (LTP) oxidation as a pretreatment method to improve the leaching ability of rare earth elements (REEs) from coal and its byproducts.
Yujian Zhou demonstrated the very effective separation of Ce3+ and La3+ by in situ electrochemical redox, which combines extraction and stripping. A new concept of combining the in situ electrolysis oxidation/extraction (IEOE)and in situ electro reduction stripping (IERS)processes was proposed.
The loaded organic phase (OP)containing D2EHPA/TBP is readily stripped by 0.5 M H2SO4, which can be used without further treatment in the developed CeRES process to obtain the individual REE ions in high purity in a single step without the need of additional reagents.
Reviewer 3 Report
Comments and Suggestions for Authors
Summary of the research progress on advanced engineering, processes and process parameters of rare earth green metallurgy is very important and excellent review paper. Minor improvements are required.
Line 13: (precipitation methods, microemulsion methods, roasting-sulfuric acid leaching methods, and electrochemical methods). Purification or separation method such as solvent extraction is missing.
Line 21: and the (molten salt) electrolysis method is a more specific method
Line 57: The emergence of electrolysis methods combined with positive-ion membrane exchange technology has improved this problem. Molten salt electrolysis produces CF4 , chlorine and fluorine. How to explain it concerning to green metallurgy.
Line 62: Green metallurgy of rare earth elements is needed to achieve the goals of short processing, low energy (and chemicals) consumption and low rare earth emission (“low CO2, fluorine and chlorine emission?).
Line 84; The roasting-sulfuric acid leaching method mainly uses rare earth ore as the raw material. Which type of minerals (bastnaesite, eudialyte, xenotime, monazite?) were used?
Line 85; When the reaction temperature is high (leaching at an atmospheric pressure (T <100°C), leaching at high pressure conditions (T <250°C))
Line 111 crystalline rare earth carbonat. (crystalline rare earth carbonate)
Line 177: What is full name for AMD at Figure 3.
Line 341, 342: The ore samples were subjected to high-pressure hydrochloric acid leaching with 10%~20% HCl, followed by magnetic separation of metallic iron. At which temperature?
Line 375; Schematic diagram of the mechanism of extraction of REEs and recovery of by-product via 375 mechanochemical processand /(processes and..)
Line 377: 2.4, Electrolysis (Electrolysis and solvent extraction)
Line 402: Prakash Venkatesan used electrochemistry to selectively extract REEs from NdFeB magnet waste at room temperature. What is an electrolyte in this case? How to solve big problem with impurities (Boron,..)
Line 555: (Molten salt) Electrolysis is (“might become) a green and environmentally friendly production process.
Author Response
Manuscript Materials-3089342
- Response to Reviewers
Dear Editor and Reviewers,
Thank you for giving us the opportunity to submit a revised draft of the manuscript “Summary of the research progress on advanced engineering, processes and process parameters of rare earth green metallurgy” for publication in the Materials. We appreciate the time and effort that you and the reviewers dedicated to providing feedback on our manuscript and are grateful for the insightful comments on and valuable improvements to our paper.
Reviewer 3:
A summary of the progress of research on advanced engineering, processes and process parameters for green metallurgy of rare earths is a very important and excellent review paper. It needs a little improvement.
We would like to thank the reviewers for recognizing our manuscript, which has received the comment “The summary of advanced engineering, process and process parameter research progress in rare earth green metallurgy is a very important and excellent review paper.” We are very pleased with this evaluation. At the same time, we would like to express our gratitude to the reviewers for the time and effort they have put into the manuscript. The suggestions made by the reviewers are very correct, and we have revised the manuscript according to the suggestions and marked them in blue font. The following is a point-by-point response to the reviewers' suggestions, which we hope you will find satisfactory.
- Line 13: (precipitation methods, microemulsion methods, roasting-sulfuric acid leaching methods, and electrochemical methods). Purification or separation method such as solvent extraction is missing.
The suggestions made by the reviewers are very correct, and we have added purification or separation methods such as solvent extraction, and we have listed the changes below for your convenience.
To this end, this paper reviews mainstream rare earth production processes (precipitation methods, microemulsion methods, roasting-sulfuric acid leaching methods, and electrochemical methods, Solvent extraction methods, ion exchange methods) to provide basic information for the green smelting of rare earth metals and help promote the development of green rare earth smelting.
2.Line 21: and the (molten salt) electrolysis method is a more specific method
The suggestions made by the reviewers are very sound, and we have described the methodology here in a normalized way based on your suggestions, and we have listed the changes below for your convenience.
the roasting-sulfuric acid leaching method is mostly used for the treatment of raw rare earth ores; and the molten salt electrolysis method is a more specific method.
- Line 57: The emergence of electrolysis methods combined with positive-ion membrane exchange technology has improved this problem. Molten salt electrolysis produces CF4, chlorine and fluorine. How to explain it concerning to green metallurgy.
The suggestions made by the reviewers are very correct, and we have described the relevance of the processes belonging here to green metallurgy according to your suggestions, and we have listed the changes below for your convenience.
The electrolysis method we mentioned in the manuscript is not molten salt electrolysis in the traditional sense, but a new electrolytic conversion method, which refers to the method of electrolytic preparation of NaOH in the chlor-alkali industry, and utilizes an electrolytic cell to electrolyze chlorinated rare earth solutions to produce rare earth compounds, which uses a cation-exchange membrane to separate the cathode chamber and the anode chamber of the electrolytic cell, and under the driving effect of the electric field, the rare earth cations pass through the cation-exchange membrane Under the driving force of an electric field, rare earth cations pass through the cation exchange membrane and enter the cathode chamber and precipitate to obtain rare earth compounds. This process does not produce CF4 and is a green process.
- Line 62: Green metallurgy of rare earth elements is needed to achieve the goals of short processing, low energy (and chemicals) consumption and low rare earth emission (“low CO2, fluorine and chlorine emission?).
The reviewer's suggestion is spot on, and we have described the process described here as low CO2, fluorine, and chlorine emissions as you suggested, and we have included the changes below for your convenience.。
Rare earth green metallurgy short process is to reduce energy consumption and human consumption, reduce the loss of heat energy in the production process, less emission is to make improvements in the process to reduce the emission of wastewater containing ammonia and nitrogen, sulfur-containing and fluorine-containing gases.
- Line 84; The roasting-sulfuric acid leaching method mainly uses rare earth ore as the raw material. Which type of minerals (bastnaesite, eudialyte, xenotime, monazite?) were used?
The suggestions made by the reviewer are very correct, and we have added the mineral types here as you suggested, and we have listed the changes below for your convenience.
The roasting-sulfuric acid leaching method mainly uses rare earth ore as the raw material, such as cerium fluorocarbonate, monazite, yttrium phosphorite.
- Line 85; When the reaction temperature is high (leaching at an atmospheric pressure (T <100°C), leaching at high pressure conditions (T <250°C))
The suggestions made by the reviewer are very correct, and we have added the reaction conditions here based on your suggestions, and we have listed the changes below for your convenience.
When the reaction temperature is high, leaching at an atmospheric pressure (T <100°C), leaching at high pressure conditions (T <250°C)
- Line 111 crystalline rare earth carbonat. (crystalline rare earth carbonate)
The suggestions made by the reviewers are very correct, and we have modified the crystalline rare earth carbonate here according to your suggestions, and we have listed the changes below for your convenience.
method to optimize the reactive crystallization process of rare earth carbonate and to prepare crystalline rare earth carbonate.
- Line 177: What is full name for AMD at Figure 3.
The suggestions made by the reviewer are spot on, and we have added the full AMD process name here as you suggested, and we have listed the changes below for your convenience.
Acid Mine Drainage (AMD)
- Line 341, 342: The ore samples were subjected to high-pressure hydrochloric acid leaching with 10%~20% HCl, followed by magnetic separation of metallic iron. At which temperature?
The suggestions made by the reviewer are very correct, and we have added the reaction conditions here based on your suggestions, and we have listed the changes below for your convenience.
The samples were subjected to high pressure acid leaching in the range of 50 to 250 degrees Celsius at an agitation speed of 350 rpm for a duration of 30 to 90 minutes with 10% to 20% HCl.
- Schematic diagram of the mechanism of extraction of REEs and recovery of by-product via 375 mechanochemical processand /(processes and..)
The reviewer's suggestions are spot on, and we've changed the language here based on your suggestions, which we've listed below for your convenience.
Schematic diagram of the mechanism of extraction of REEs and recovery of by-product via mechanochemical processand.
- Line 377: 2.4, Electrolysis (Electrolysis and solvent extraction)
The suggestions made by the reviewer are spot on, and we have changed the language here based on your suggestions, and we have listed the changes below for your convenience.
2.4, Electrolysis and solvent extraction
12 Line 402: Prakash Venkatesan used electrochemistry to selectively extract REEs from NdFeB magnet waste at room temperature. What is an electrolyte in this case? How to solve big problem with impurities (Boron,)
The suggestions made by the reviewers are spot on, and we will explain them in the light of your suggestions, and we have listed the changes below for your convenience.
Prakash Venkatesan used electrochemistry to selectively extract REEs from NdFeB magnet waste at room temperature.In this electrolysis pretreatment step with NH4Cl as the electrolyte, NdFeB magnet waste dissolves as an active metal anode (AMA) and simultaneously, a Ti/Pt inert anode (IA) oxidizes Fe(II) to Fe(OH)3. Iron hydroxides have the capacity to act as metal scavengers and boron is often removed from the leachate by coagulation or electrocoagulation with ferric hydroxide.
13 Line 555: (Molten salt) Electrolysis is (“might become) a green and environmentally friendly production process.
The reviewer's suggestions are spot on, and we've changed the language here based on your suggestions, which we've listed below for your convenience.
Electrolysis might become a green and environmentally friendly production process.
Round 2
Reviewer 1 Report
Comments and Suggestions for Authors
Since you've performed serious revision of the submitted manuscript both in the all sections and cited references. Current, revised version of the manuscript is publishable. Congratulations